# HIV-1 promotes ubiquitination of the amyloidogenic C-terminal fragment of APP to support viral replication

Feng Gu[1], Marie Boisjoli[1] & Mojgan H. Naghavi ®[1] ✉

HIV-1 replication in macrophages and microglia involves intracellular assembly and budding into modified subsets of multivesicular bodies (MVBs), which support both viral persistence and spread. However, the cellular factors that regulate HIV-1's vesicular replication remain poorly understood. Recently, amyloid precursor protein (APP) was identified as an inhibitor of HIV-1 replication in macrophages and microglia via an unknown mechanism. Here, we show that entry of HIV-1 Gag into MVBs is blocked by the amyloidogenic C-terminal fragment of APP, "C99", but not by the non-amyloidogenic product, "C83". To counter this, Gag promotes multi-site ubiquitination of C99 which controls both exocytic sorting of MVBs and further processing of C99 into toxic amyloids. Processing of C99, entry of Gag into MVBs and release of infectious virus could be suppressed by expressing ubiquitination-defective C99 or by γ-secretase inhibitor treatment, suggesting that APP's amyloidogenic pathway functions to sense and suppress HIV-1 replication in macrophages and microglia.

Human Immunodeficiency Virus type 1 (HIV-1) infects a number of immune cell types including CD4+ T-cells, macrophages and brain-resident microglia, which underlies its complex pathophysiology that includes immune cell depletion as well as neurological disorders[1]. Beyond differential co-receptor usage during virus entry, the late-stage intracellular replication and production of infectious HIV-1 particles is complex and ultimately highly cell-type dependent. Virion assembly involves multimerization of the Gag (Group-specific antigens) polyprotein at cellular membranes; membrane association is driven by myristoylation and both phospholipid and protein interactions mediated by its matrix (MA) domain, multimerization and RNA packaging is driven by domains in capsid (CA) and nucleocapsid (NC), and recruitment of class E vacuolar protein sorting (VPS) proteins is mediated by its C-terminal (p6) domain[2–4]. The VPS machinery drives viral budding by invaginating regions of the plasma membrane or vesicular membranes, where Gag is then processed to form a mature virion surrounded by a host-derived lipid envelope containing the viral Envelope (Env) protein. Use of the VPS machinery along with visualization of Gag and mature virions within intracellular vesicles initially

led to a model for HIV-1 assembly within a subset of modified late endosomes (LE)- or multivesicular bodies (MVBs), often termed virus-containing compartments (VCCs)[5]. Indeed, MVBs are used by several viruses for assembly and release[6,7]. However, it is now clear that in T-cells Gag is predominantly localized to the plasma membrane which acts as the primary site of budding, recruiting what is normally viewed as vesicular VPS machinery[4,8–10]. But in "professional phagocytes" such as macrophages and microglia the situation is more complex. In these cells, while there is some degree of plasma membrane-based assembly during the initial stages of replication, Gag and mature virions are predominantly found within intraluminal vesicles of MVBs that appear linked to MHC Class II secretion and express distinguishing markers such as the tetraspanin, CD63[11]. Moreover, after budding into these sites, CD63+ MVBs or VCCs accumulate at the cell surface and play a key role in cell-cell spread by macrophages or microglia, as well as potentially acting as reservoirs of infectious virus in these cell types. These intracellular/intraluminal localization and assembly phenotypes are recapitulated in several cell lines including HEK293 cells[10,12–15]. Indeed, the widespread use of cell lines likely contributed to initial

[1]Department of Microbiology-Immunology, Northwestern University Feinberg School of Medicine, Chicago, IL, USA.
✉e-mail: mojgan.naghavi@northwestern.edu

models of intracellular/intraluminal assembly and as such, it seems likely that HEK293 and other transformed cell types better model late-stage events in macrophages/microglia than virus replication in T-cells. However, while budding from the plasma membrane in T-cells is relatively well defined, the host factors and processes involved in the intracellular assembly and trafficking of HIV-1 in macrophages/microglia remains poorly understood.

We recently found that amyloid precursor protein (APP) is highly expressed in macrophages and microglia and suppresses HIV-1 replication[16]. Moreover, to evade this restriction, HIV-1 Gag increases APP processing but in doing so, infection increases the production of neurotoxic APP-derived amyloids. APP is a transmembrane protein with three major isoforms; $APP_{695}$ is mainly expressed in neurons while $APP_{751}$ and $APP_{770}$ are the predominant isoforms expressed in non-neuronal cells[17]. Localizing to both the plasma membrane and intracellular vesicles, APP consists of a large extracellular domain, a single trans-membrane domain and a short cytoplasmic tail. While APP is best known as the precursor of toxic β-amyloid (Aβ) peptides, its biological function is surprisingly less well-understood but neurons lacking APP exhibit impaired intracellular trafficking, neurite growth and cell-to-cell adhesion[18]. This suggests that APP regulates trafficking but why it evolved to produce toxic or inflammatory amyloids remains unclear. APP is processed by distinct secretases through two proteolytic pathways; amyloidogenic versus non-amyloidogenic (although another non-amyloidogenic pathway exists for neuron-specific $APP_{695}$) (Fig. 1a). The non-amyloidogenic pathway is more frequent during normal homeostasis wherein APP is cleaved by α-secretase at the plasma membrane to yield the membrane tethered C-terminal fragment (α-CTF or "C83") while releasing soluble APP fragment alpha (sAPPα) into the extracellular space. Further processing of C83 by γ-secretase releases a non-toxic secretory peptide (p3) and a cytosolic APP intracellular domain (AICD). In the amyloidogenic pathway, which occurs predominantly in endosomal compartments, APP is first endocytosed and cleaved by β-secretase to generate a soluble ecto-domain (sAPPβ) and a C-terminal fragment (β-CTF or "C99"). γ-secretase further processes C99 into Aβ peptides of various lengths (39–43 aa) as well as AICD. The localization and fate of amyloidogenic APP fragments is complex and determined by intracellular sorting of late endosomes to either lysosomes for clearance during normal homeostasis, or to exocytic multivesicular bodies (MVBs) that release toxic amyloids under pathological conditions. While the biological purpose of making Aβ peptides remains unclear, their accumulation within neurons and their release into the brain environment by neurons or brain-resident immune cells results in the formation of plaques and inflammation that is associated with dementia[18,19]. Although Aβ levels in the brain increase during normal aging, this is greatly accelerated by HIV-1 infection and correlates with viral loads and the onset of HIV-1 associated neurodegenerative disorders (HAND)[20]. Multiple aspects of HIV-1 infection and even antiretroviral therapy itself contribute to the complex inflammatory environment that broadly alters APP metabolism in the brain[21]. But the discovery that HIV-1 evades restriction by APP by increasing its amyloidogenic processing offers both an explanation for why infection increases toxic amyloid production and a therapeutic opportunity to simultaneously block amyloid production and harness APP's antiviral activity through the use of γ-secretase inhibitors[16]. However, the mechanistic basis by which APP inhibits HIV-1 replication remains undefined.

Here, we reveal that HIV-1 promotes the multi-site ubiquitination and degradation of the C-terminal C99 product derived from amyloidogenic processing of APP, but not the C83 product derived from non-amyloidogenic processing pathways. While C83 does not inhibit infection, blocking C99 degradation using ubiquitin-site mutants or preventing its further processing into toxic β-amyloids using a γ-secretase inhibitor impairs Gag entry into CD63-positive MVBs and suppresses the release of infectious viral particles. Combined, our data

suggests that an important but unrecognized function of the branched amyloid processing pathway of APP is to impair the use of MVBs by pathogens such as HIV-1, evasion of which triggers the release of amyloids that may in turn act as inflammatory pathogen alerts within the brain environment.

## Results

### HIV-1 Gag increases processing of APP and CTFs

While our prior work established that HIV-1 increased APP processing to evade its antiviral effects, how infection impacts amyloidogenic versus non-amyloidogenic CTFs and the underlying mechanism by which APP inhibits infection remain unknown. To first determine its effects on CTFs that are downstream products of APP processing, we transfected HEK293A cells with plasmids encoding the full-length HIV-1 infectious clones of X4-tropic pNL4.3 or R5-tropic JR-CSF. Western blotting (WB) revealed that infection with either virus strain caused a decrease in both full-length APP and its CTF products (Fig. 1b). Demonstrating the relevance of findings using HEK293 cell systems, infection of CHME3 4×4 (a human microglia line which expresses higher levels of CD4 and CXCR4 for more efficient infection with wildtype HIV-1 envelope)[16,22] or primary human monocyte-derived macrophages (MDMs) with either X4-tropic pNL4.3 or R5-tropic JR-CSF resulted in a similar decrease in both full-length APP and its CTFs (Fig. 1c, d, respectively). Our previous work identified a region spanning amino acids 72-111 in the C-terminus of MA domain that influences Gag membrane localization and is required for HIV-1 Gag to increase APP processing[16]. Interestingly, others have identified more refined mutations within the same region, between amino acids 84 and 88 in MA, that also alter Gag membrane usage[23]. Transfecting HEK293A cells with two of these mutants, namely 85YG and 87VE in the pNL4.3 background[23] revealed that they also failed to reduce both APP and CTFs (Fig. 1e). These findings provide further evidence that the MA region plays a critical role in HIV-induced APP processing, which data below suggests is due to competition for intracellular vesicles that support virus replication and APP processing.

The ability of HIV-1 to downregulate APP and CTFs was reduced by treating cells with Bafilomycin A1 (BafA1) (Fig. 1f), in line with the studies establishing that amyloidogenic processing of APP and its CTF, C99 occurs within endosomal compartments[24–30]. While we have shown previously that expression of HA-tagged Gag is sufficient to downregulate APP[16], extending on this we find that transfection of HEK293A cells with Gag-HA is also sufficient to accelerate CTF processing (Fig. 1g). In line with this, increasing amounts of Gag-HA, but not GAPDH-HA, resulted in a reduction in both APP and CTFs in the presence of cycloheximide (CHX) to prevent de novo APP synthesis (Fig. 1h). The fact that CTFs do not accumulate despite increased upstream cleavage of APP demonstrates that HIV-1 infection or HIV-1 Gag alone accelerate the entire pathway, and not just APP cleavage alone. Amyloidogenic processing of APP involves a combination of vesicular sorting that is sensitive to BafA1 treatment along with proteasomal degradation that defines the level of amyloid clearance versus secretion[24–30]. In line with this, treatment with BafA1 significantly reduced processing of full-length APP while both BafA1 and the proteasome inhibitor, MG132, suppressed the subsequent downregulation of CTFs in Gag-expressing cells (Fig. 1g). This suggests that HIV-1 exploits vesicular processing pathways that branches at least in part, to proteasome clearance versus exocytosis. Notably, BafA1 or MG132 treatment also affected the basal levels of APP and CTFs in control samples (Fig. 1f, g), which would be expected due to disruption of the ongoing basal level of APP processing and clearance in these cells. However, our data demonstrates that HIV-1 accelerates these vesicle-based amyloidogenic processing and sorting pathways. Further supporting this idea, the γ-secretase inhibitor L685,458 blocked the cleavage of

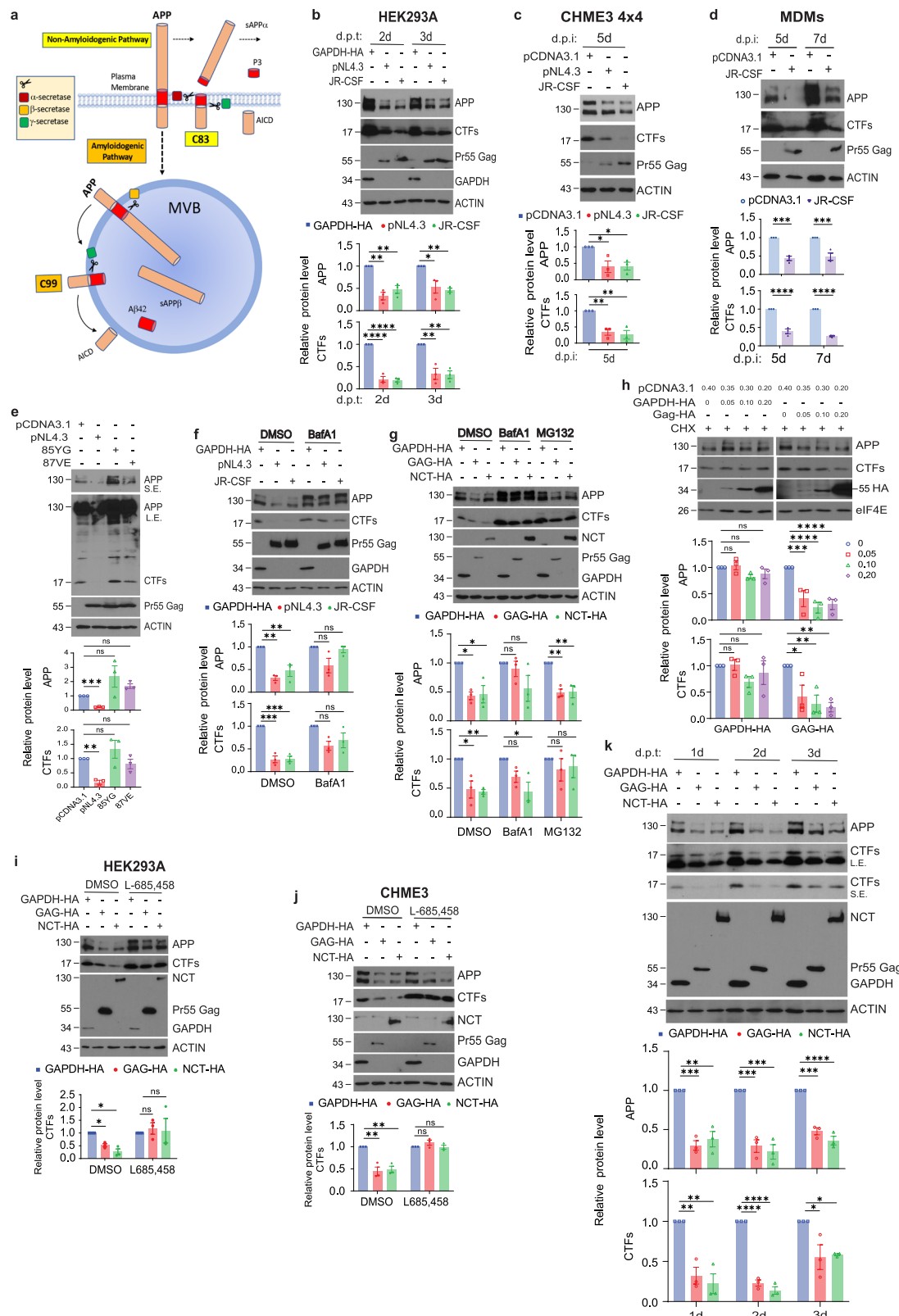

CTFs but not the upstream cleavage of APP by β-secretase, as would be expected, in Gag-transfected HEK293A cells or CHME3 microglia (Fig. 1i, j). Interestingly, the effects of Gag transfection were similar to those of transfecting Nicastrin (Fig. 1g, i, k), the APP- and membrane-associated host protein that drives amyloidogenic processing[31], suggesting that Gag may functionally mimic Nicastrin in order to increase APP turnover.

## HIV-1 ubiquitinates and degrades C99, but not C83

While our data established that HIV-1 infection or Gag expression accelerated vesicle- and γ-secretase-based processing of APP and its CTFs, available antibodies do not reliably distinguish between C99 or C83 given their high degree of homology. To address this, we expressed N-terminally Flag-tagged forms of non-amyloidogenic C83 or amyloidogenic C99 in HEK293A or CHME3 cells. Imaging showed

**Fig. 1 | HIV-1 Gag increases processing of APP and CTFs. a** APP processing pathways. During non-amyloidogenic processing APP is cleaved at the plasma membrane by α-secretase to produce external sAPPα and C83, followed by γ-secretase resulting in p3 and AICD fragments. During amyloidogenic processing APP is endocytosed and trafficked to late endosomes/multivesicular bodies (MVBs), where it is cleaved by β-secretase to produce sAPPβ and C99, followed by γ-secretase cleavage into Aβ42 and AICD. Transfection of HEK293A (**b**) or infection of CHME3 4 × 4 (**c**) or MDMs (**d**) with HIV-1 pNL4.3 or JR-CSF downregulates APP and CTFs at the indicated days post transfection (d.p.t) or infection (d.p.i). **e** pNL4.3, but not pNL4.3 containing point mutations in MA domain (85YG and 87VE), results in a reduction in both APP and CTFs in transfected HEK293A cells. L.E., long exposure; S.E., short exposure. Bafilomycin A1 (BafA1) or MG132, but not DMSO control treatment prevents APP and CTF downregulation by transfected pNL4.3 or JR-CSF (**f**) or HA-tagged Gag (Gag-HA) or Nicastrin (NCT-HA) (**g**) in

HEK293A cells. **h** Increasing amount of Gag-HA, but not GAPDH-HA, results in a reduction in both APP and CTFs in transfected HEK293A cells treated with cycloheximide (CHX). GSI L685,458 inhibits both basal CTF turnover in untransfected cells and accelerated turnover in Gag-HA- or NCT-HA-transfected HEK293A (**i**) or CHME3 (**j**) cells. **k** Timecourse transfection of HEK293A cells with Gag-HA or NCT-HA, but not with the control GAPDH-HA, reduces the levels of APP and CTF at the indicated d.p.t. Quantification of the protein levels from 3 independent replicates is presented below each panel shown in **b**–**k**. Data is presented as mean, SEM using one-way ANOVA with Tukey's multiple comparisons test (in **b**, **c**, **f**, **g**, **j**, and **k**), two-way ANOVA with Sidak's multiple comparisons test (in **d**, **h** APP graph) or Dunnett's multiple comparisons test (in **h** CTFs graph); two tailed one sample $t$ test (in **e**); unpaired two tailed $t$ test with Welch's correction (in **i**). *$p < 0.05$, **$p < 0.01$, ***$p < 0.001$, ****$p < 0.0001$, ns: not significant. Source data are provided as a Source Data file.

that similar to endogenous APP detected with the LN27 antibody and in line with amyloidogenic processing within intracellular vesicles, Flag-tagged C99 localizes to both early and late endosomes (EEA1- and Rab7-positive vesicles, respectively), with a more robust accumulation in the latter compartment in CHME3 cells (Supplementary Fig. 1a, b, d, e, and g, h). By contrast, Flag-tagged C83 was more readily detected at both the plasma membrane and within endocytic compartments, in line with non-amyloidogenic processing occurring primarily at the plasma membrane (Supplementary Fig. 1c, f, and i).

Using these constructs to determine effects of infection on amyloidogenic versus non-amyloidogenic pathways, we first co-transfected HEK293A cells with GAPDH control, Flag-C99 or Flag-C83 together with increasing amounts of HIV-1 JR-CSF. Results demonstrated that increasing amount of JR-CSF resulted in a corresponding dose-dependent decrease in the levels of exogenous C99, but not in the levels of exogenously expressed C83 or control GAPDH (Fig. 2a, b). Ubiquitination has been shown to play a key role in regulating the sorting and amyloidogenic processing of C99[24–30]. Testing this, we found that the ubiquitination inhibitor, TAK-243 specifically stabilized exogenous C99 but not C83 in HEK293A cells transfected with either empty vector control or JR-CSF (Fig. 2c, d). In line with our earlier BafA1 data above, TAK-243 caused an increase in the basal levels of exogenous C99 in control samples. In line with ubiquitin-mediated turnover of C99, treatment with MG132 or TAK-243 (Fig. 2e, f) or RNAi-mediated depletion of ubiquitin-activating enzyme E1 (UBE1) (Fig. 2g, h) prevented the downregulation of exogenous C99 in JR-CSF transfected HEK293A cells. Combined, these results further support the notion that HIV-1 accelerates the basal amyloidogenic processing of C99 through ubiquitination.

The C99 region of APP contains 7 lysines that are potential ubiquitination sites, 3 of which cluster at K724-726 (Fig. 2i). Studies have shown that mutation of K724-726 impairs APP ubiquitination by 75% and blocks recruitment of ESCRT (endosomal sorting complexes required for transport) proteins that mediate APP/C99 insertion into MVBs[26,28]. Notably, this blocks sorting to and processing within exocytic pathways that increase amyloid secretion. In line with these findings, lysine to alanine (K-A) mutagenesis showed that while individual lysines contribute in a small way to C99 stability, mutation of K724-726 substantially restored C99 stability while mutation of all 7 lysines (7KA mutant) recapitulated the stabilization effects of TAK-243 (Fig. 2j, k). Furthermore, anti-Flag immunoprecipitation (IP) analysis of lysates from cells expressing Flag control, Flag-C99 or Flag-C99-7KA that were transfected with control or HIV-1 JR-CSF plasmid again showed downregulation of C99 and stabilization by the 7KA mutation in input samples (note, concentration of C99 in IP'd samples along with longer exposures required to visualize input C99 mask its downregulation in bound samples) (Fig. 2l). Probing bound samples for ubiquitin (Ub) also showed several specific Ub-reactive bands in C99 IP's from JR-CSF-transfected cells that were not readily detected above background levels in the 7KA mutant. While the 7KA mutant or TAK-

243 also prevents C99 turnover in control Flag-transfected cells, these results further show that HIV-1 infection increases C99 ubiquitination and increases its processing. Furthermore, when combined with the inhibitory effects of BafA1 treatment, these data suggest that HIV-1 tilts the balance of amyloidogenic processing away from lysosomal clearance and towards exocytic pathways[26,28,32], which would explain our prior observation of increased amyloid secretion by infected cells[16] and would align with the broader notion that HIV-1 promotes exocytic sorting of MVBs as a means to traffic to the cell surface to spread.

### Preventing C99 ubiquitination suppresses HIV-1 Gag entry into CD63-positive compartments/vesicles

To explore this further, we first determined whether HIV-1 infection broadly overlaps with and affects APP sorting pathways by infecting CHME3 cells with either pNL4.3- or JR-CSF-derived HIV-1. Fixed samples were then stained for Gag, APP (detected using LN27 anti-APP antibody[27,33]) and LAMP1. Imaging revealed a complex set of colocalization patterns (Supplementary Fig. 2); some vesicles containing mixtures of Gag and/or APP stained strongly for LAMP1, which is characteristic of lysosomes that are involved in clearance of APP but are inhibitory to HIV-1[34]. Others that contained Gag alone or both Gag and APP stained weakly for LAMP1, which is a characteristic of exocytic MVBs that support infection[11,35]. This supports the broader idea that Gag and APP intersect at vesicular sorting pathways, competition for control of which likely dictates the balance of virus replication and amyloid clearance versus secretion.

The balance of lysosomal and secretory processing of APP is regulated by ubiquitination of its C-terminus, which promotes engagement with the VPS machinery and routing to exocytic vesicles[26,28,32]. This suggests that increased C99 ubiquitination (Fig. 2l) might be important for HIV-1 to control this balance and promote exocytic sorting. To test this, we next co-transfected HEK293A cells with the infectious clone JR-CSF together with either Flag-C99 or Flag-C99-7KA. Staining samples for the LE marker, Rab7 revealed that in cells expressing Flag-C99, Gag formed typical punctate structures that frequently colocalized with Rab7-positive vesicles (Fig. 3a–c; note, due to degradation C99 is overexposed to detect expressing cells and localization patterns). By contrast, in cells expressing Flag-C99-7KA Gag failed to localize to LE/MVBs and instead exhibited a diffuse plasma membrane or cytosolic localization pattern. Similar results were obtained in JR-CSF transfected in CHME3 microglia cells expressing either Flag-C99 or Flag-C99-7KA (Fig. 3d–f). As C99 showed no obvious defects in localizing to Rab7 compartments and based on defects in vesicular budding by ubiquitin mutants of APP[26,28], these findings suggest that C99-7KA limits Gag's access to MVBs. Moreover, imaging also showed that C99-7KA localized more strongly than C99 to EEA1-positive early endosomes (Supplementary Fig. 3a, b), suggesting that the inhibition of C99 ubiquitination likely affects endosomal sorting pathways.

While Rab7 acts a convenient, general marker for LE's, CD63 is well established as a specific marker of the MVB subsets that support HIV-1

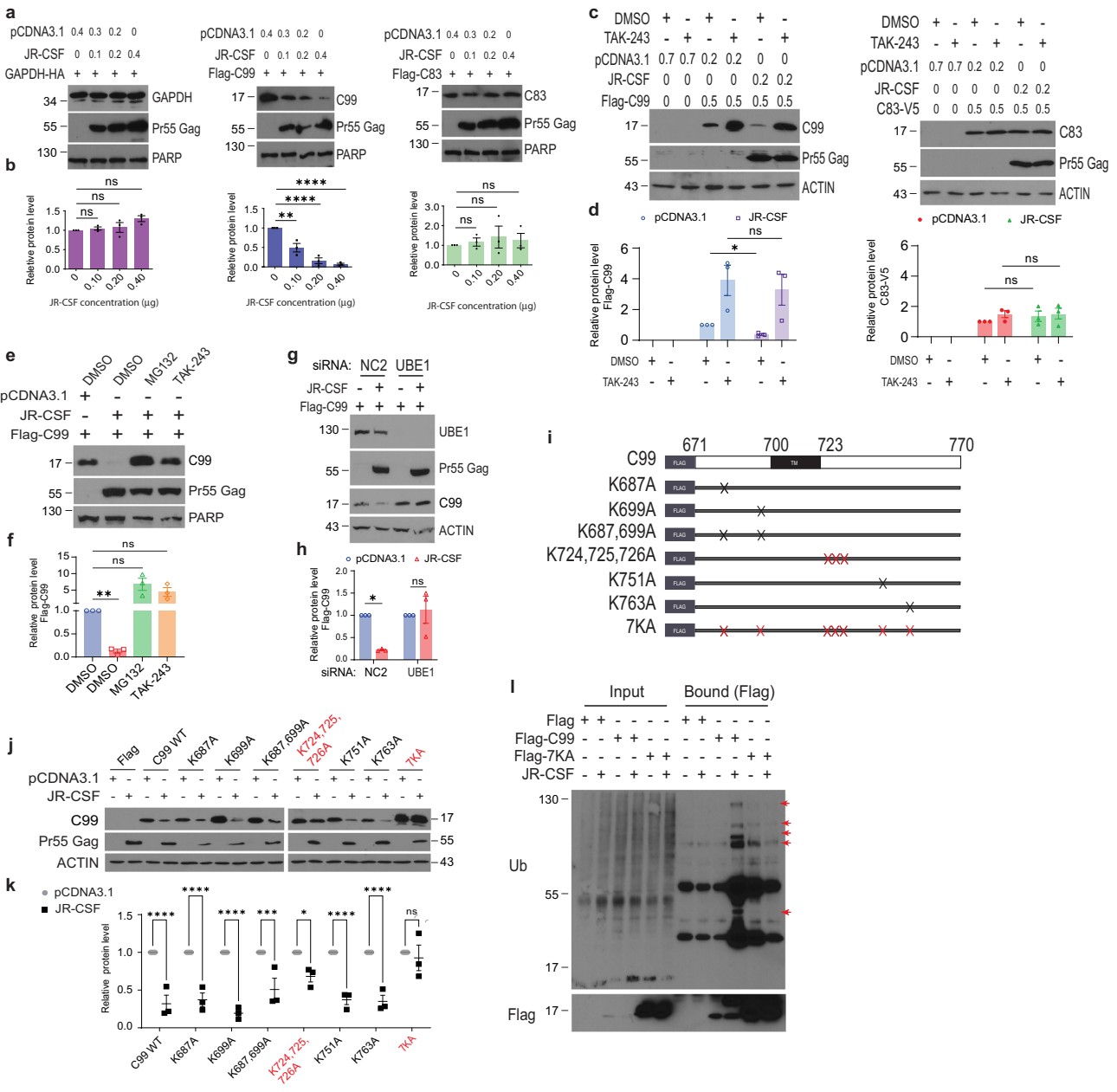

**Fig. 2 | HIV-1 ubiquitinates and degrades C99, but not C83. a** Transfection of HIV-1 JR-CSF reduces Flag-C99, but not Flag-C83 or GAPDH-HA in HEK293A cells. **b** Densitometry analysis of effects in (**a**) from 3 independent replicates. **c** Ubiquitination inhibitor TAK-243 stabilizes exogenous Flag-C99 but not C83-V5 in HEK293A cells transfected with either empty vector control (pcDNA3.1) or JR-CSF. **d** Densitometry analysis of effects in **c** from 3 independent replicates. **e** MG132 and TAK-243, but not DMSO control, stabilizes exogenous Flag-C99 in HEK293A cells transfected with JR-CSF. **f** Densitometry analysis of effects in (**e**) from 3 independent replicates. **g** Ubiquitin-activating enzyme E1 (UBE1) specific siRNA, but not negative control siRNA (NC2), suppresses downregulation of exogenous Flag-C99 in JR-CSF transfected HEK293A cells. **h** Densitometry analysis of effects in (**g**) from 3 independent replicates. **i** Schematic of N-terminally Flag-tagged WT or mutant C99 constructs used in (**j**). K: Lysine; A. Alanine. Each mutation site at the corresponding amino acid is represented by X. WB (**j**) and densitometry analysis (**k**) of effects of single or combined lysine-alanine (K-A) mutations on Flag-C99 stability in control pcDNA3.1 or JR-CSF transfected HEK293A cells. Complete stabilization mirroring TAK-243 treatment occurs in the C99-7KA mutant. **l** Representative (n = 3) anti-Flag IP showing ubiquitination of Flag-C99 but not Flag-C99-7KA mutant in lysates from HEK293A transfected with JR-CSF. Red arrows highlight ubiquitinated bands. Quantification of the protein levels from 3 independent replicates is presented in **b**, **d**, **f**, **k**. Data is presented as mean, SEM. **b** one-way ANOVA with Tukey's multiple comparisons test; **d** unpaired two-tailed $t$ test with Welch's correction; **f** unpaired two-tailed $t$ test with Welch's correction; **h** and **k**: two-way ANOVA with Sidak's multiple comparisons test. *$p < 0.05$, **$p < 0.01$, ***$p < 0.001$, ****$p < 0.0001$, ns: not significant. Source data are provided as a Source Data file.

replication[5,9,13,14,34–38]. As such, we next determined if C99 or the C99-7KA mutant affected Gag colocalization with CD63-positive vesicles. In HEK293A cells co-transfected with Flag-C99 and HIV-1 JR-CSF, Gag again formed typical punctate structures that frequently colocalized with CD63 and Flag-C99 (Fig. 4a–c). This further showed that Gag and C99 intersect at the specific MVB subsets that support HIV-1

replication in several cell types. In line with our findings using Rab7 to stain LE's, in cells expressing the ubiquitin-site mutant Flag-C99-7KA, Gag failed to localize to CD63-positive vesicles and instead exhibited a diffuse cytosolic localization (Fig. 4a–c). Similar results were obtained in JR-CSF-transfected CHME3 cells or MDMs expressing either Flag-C99 or Flag-C99-7KA (Fig. 4d–f and Fig. 5a–c, respectively). Moreover,

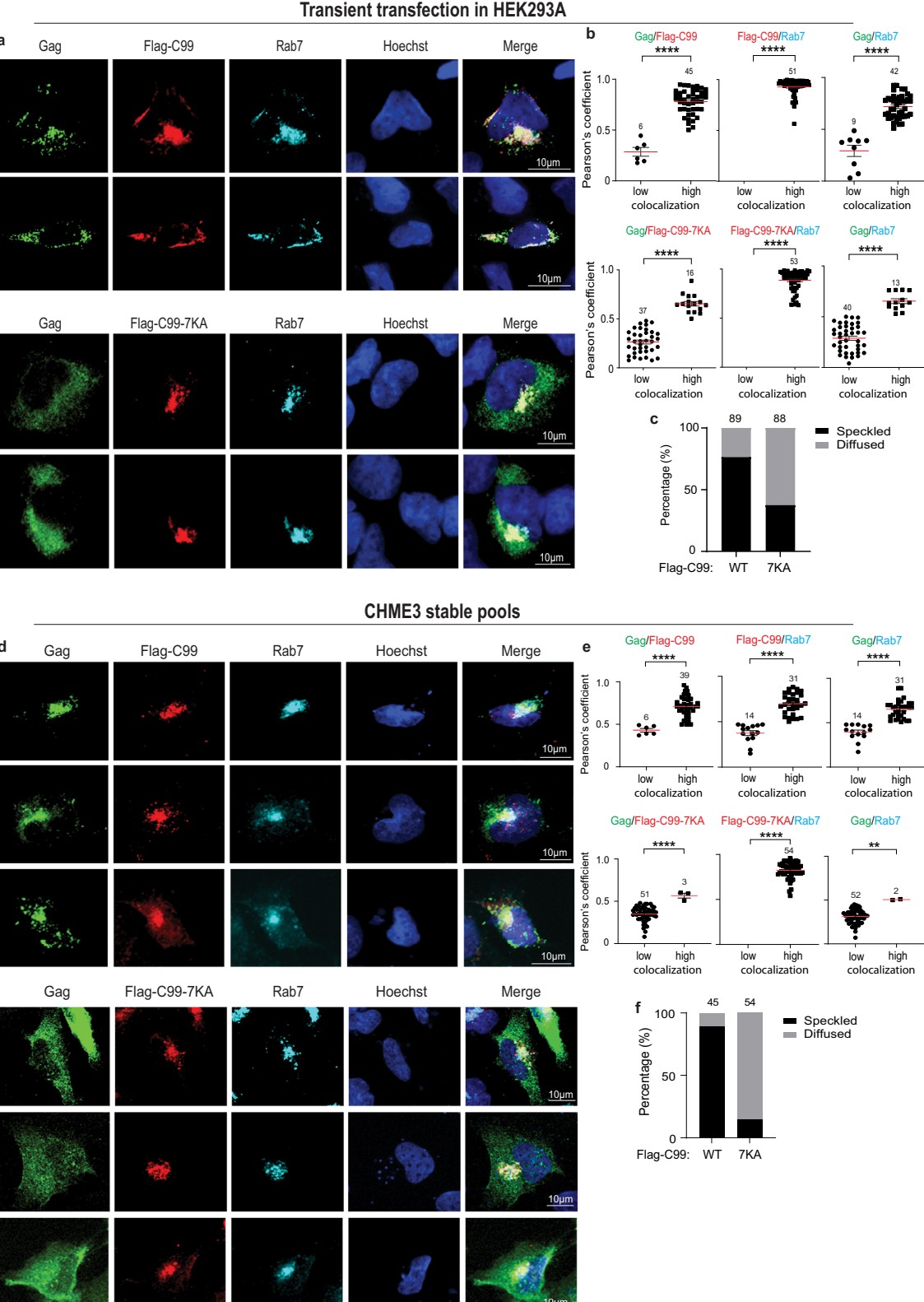

**Fig. 3 | Preventing C99 ubiquitination suppresses HIV-1 Gag entry into Rab7-positive vesicles.** JR-CSF-transfected HEK293A (**a**–**c**) or CHME3 (**d**–**f**) cells transiently or stably expressing Flag-C99 WT or Flag-C99-7KA mutant, respectively. Cells were fixed and stained for Gag, Flag and Rab7, detecting nuclei using Hoechst. Representative images of cells expressing Flag-C99 or Flag-C99-7KA in HEK293A at 48 h (**a**) or in CHME3 at 24 h (**d**) post-transfection. Quantification of the colocalization of Gag and C99, C99 and Rab7, or Gag and Rab7 under each condition in HEK293A (**b**) or CHME3 (**e**) cells determined by Pearson's Correlation Coefficient.

Data is presented as mean with SEM using two-tailed one sample Wilcoxon test with hypothetical value 0.50 (Flag-C99/Rab7, Flag-C99-7KA/Rab7 in **b**; Flag-C99-7KA/Rab7 in **e** or unpaired two-tailed $t$ test for the remaining graphs, $**p < 0.01$, $****p < 0.0001$. Number of cells analyzed is indicated. Quantification of HEK293A (**c**) or CHME3 (**f**) cells exhibiting typical punctate/speckled versus diffuse Gag distribution patterns. Number of cells analyzed is indicated. Source data are provided as a Source Data file.

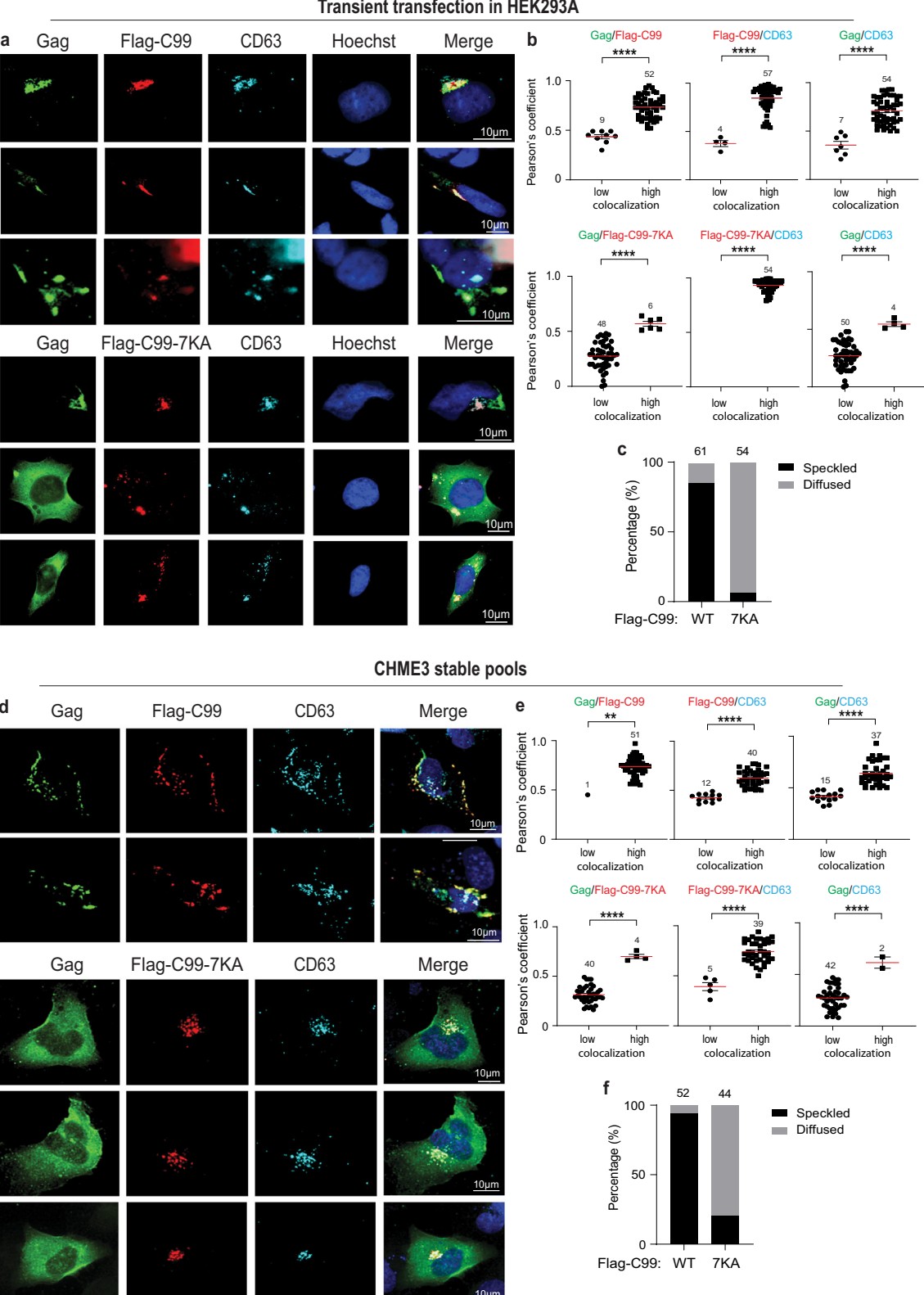

despite some degree of localization to LE's and CD63-positive MVBs, expression of Flag-tagged non-amyloidogenic C83 had no effect on Gag localization to these compartments (Fig. S3c, d). Combined, these findings in two distinct cell systems, including biologically relevant microglia and primary macrophages, suggest that the C99 cleavage product of APP is sufficient to impair HIV-1 Gag entry into CD63-positive LE/MVBs and in order to counter this, HIV-1 stimulates C99

ubiquitination and amyloidogenic processing through exocytic pathways.

### Preventing endogenous APP processing limits HIV-1 Gag entry into MVBs

We next tested whether preventing the amyloidogenic processing of endogenous APP would also block Gag entry into LE/MVBs,

**Fig. 4 | Preventing C99 ubiquitination suppresses HIV-1 Gag entry into CD63-positive vesicles.** JR-CSF-transfected HEK293A (**a**–**c**) or CHME3 (**d**–**f**) cells transiently or stably expressing Flag-C99 WT or Flag-C99-7KA mutant, respectively. Cells were fixed and stained for Gag, Flag and CD63, detecting nuclei using Hoechst. Representative images of cells expressing Flag-C99 WT or Flag-C99-7KA mutant in HEK293A at 48 h (**a**) or in CHME3 at 24 h (**d**) post-transfection. Quantification of the colocalization of Gag and C99, C99 and CD63, or Gag and CD63 under each condition in HEK293A (**b**) or CHME3 (**e**) cells determined by Pearson's Correlation Coefficient. Data is presented as mean with SEM using unpaired two-tailed $t$ test (Gag/Flag-C99 and paired Gag/CD63, Gag/Flag-C99-7KA and paired Gag/CD63 in **b**; Gag/Flag-C99, Gag/Flag-C99-7KA, Flag-C99-7KA/CD63 and paired Gag/CD63 in **e**) or unpaired two-tailed $t$ test with Welch's correction (Flag-C99/CD63 and paired Gag/CD63 in **e**) or two-tailed Mann–Whitney test (Flag-C99/CD63 in **b**) or two-tailed one sample Wilcoxon test with hypothetical value 0.50 (Flag-C99-7KA/CD63 in **b**)**$p < 0.01$, ****$p < 0.0001$. Number of cells analyzed is indicated. Quantification of HEK293A (**c**) or CHME3 (**f**) cells exhibiting typical punctate/speckled versus diffuse Gag distribution patterns. Number of cells analyzed is indicated. Source data are provided as a Source Data file.

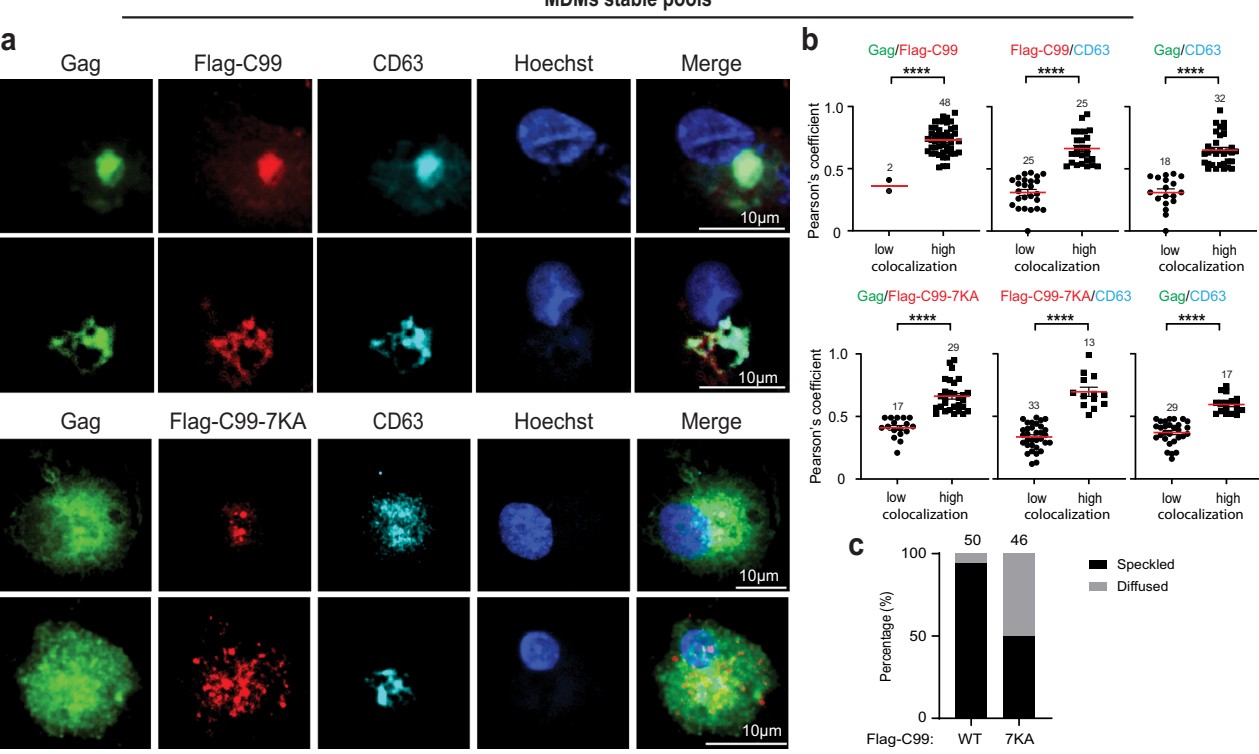

**Fig. 5 | Preventing C99 ubiquitination suppresses HIV-1 Gag entry into CD63-positive vesicles in primary human macrophages. a**–**c** MDMs stably expressing Flag-C99 WT or Flag-C99-7KA mutant were infected with JR-CSF-derived HIV-1. 5 days post infection cells were fixed and stained for Gag, Flag and CD63, detecting nuclei using Hoechst. **a** Representative images of MDMs expressing Flag-C99 WT or Flag-C99-7KA mutant are shown. **b** Quantification of the colocalization of Gag and C99, C99 and CD63, or Gag and CD63 under each condition determined by Pearson's Correlation Coefficient; mean with SEM using unpaired two-tailed $t$ test, ****$p < 0.0001$. Number of cells analyzed is indicated. **c** Quantification of cells exhibiting typical punctate/speckled versus diffuse Gag distribution patterns. Number of cells analyzed is indicated. Source data are provided as a Source Data file.

similar to our findings using the C99-7KA mutant above. To do this, we transfected HEK293A or CHME3 cells with HIV-1 JR-CSF and treated cultures with solvent control or the γ-secretase inhibitor, L685,458 (Fig. 6a–f, respectively). Although DMSO reduced the overall efficiency of Gag localization to MVBs in CHME3 cells, as expected Gag formed punctate structures that frequently localized to Rab7-positive LE/MVBs in the presence of solvent control. By contrast, Gag became diffusely distributed throughout the cytosol with very little specific localization to Rab7-positive vesicles in L685,458-treated cells. The same effects were observed in HEK293A cells expressing Flag-C99 (Supplementary Fig. 4). Moreover, no effects of L685,458 treatment on Gag localization to Rab7-positive MVBs were observed in HEK293A cells treated with siRNAs against APP (Supplementary Fig. 5), demonstrating that the effects of this γ-secretase inhibitor were mediated by its endogenous substrate, APP. To independently test these findings and their specific effects on Gag accessibility to MVBs that support virus replication, we further confirmed that L685,458 blocked Gag

localization to CD63-positive vesicles in MDMs infected with JR-CSF-derived HIV-1 (Fig. 6g–i). Testing this further, CHME3 cells or MDMs were treated with either control or APP specific siRNAs and then transfected or infected with HIV-1 JR-CSF, respectively, followed by treatment with either DMSO solvent control or L685,458 (Figs. 7 and 8, respectively). Cells were then fixed and stained for Gag and CD63 together with Hoechst. Although DMSO again reduced the overall efficiency with which Gag was internalized into MVBs in CHME3 cells, but not in MDMs, in general, the same overall results as those in HEK293A cells were observed in both cell types. In control siRNA-treated cells, L685,458 increased CTF levels, reduced the accumulation of Gag in CD63-positive vesicles, and increased the percentage of cells with diffuse Gag staining patterns (Figs. 7a–d and 8a–d, respectively). By contrast, L685,458 treatment had no significant effect on Gag localization to CD63-positive vesicles in APP-depleted CHME3 or MDMs, in line with the inability of L685,458 treatment to generate significant levels of CTFs in the absence of APP (Figs. 7a–e and 8a–e, respectively). Combined,

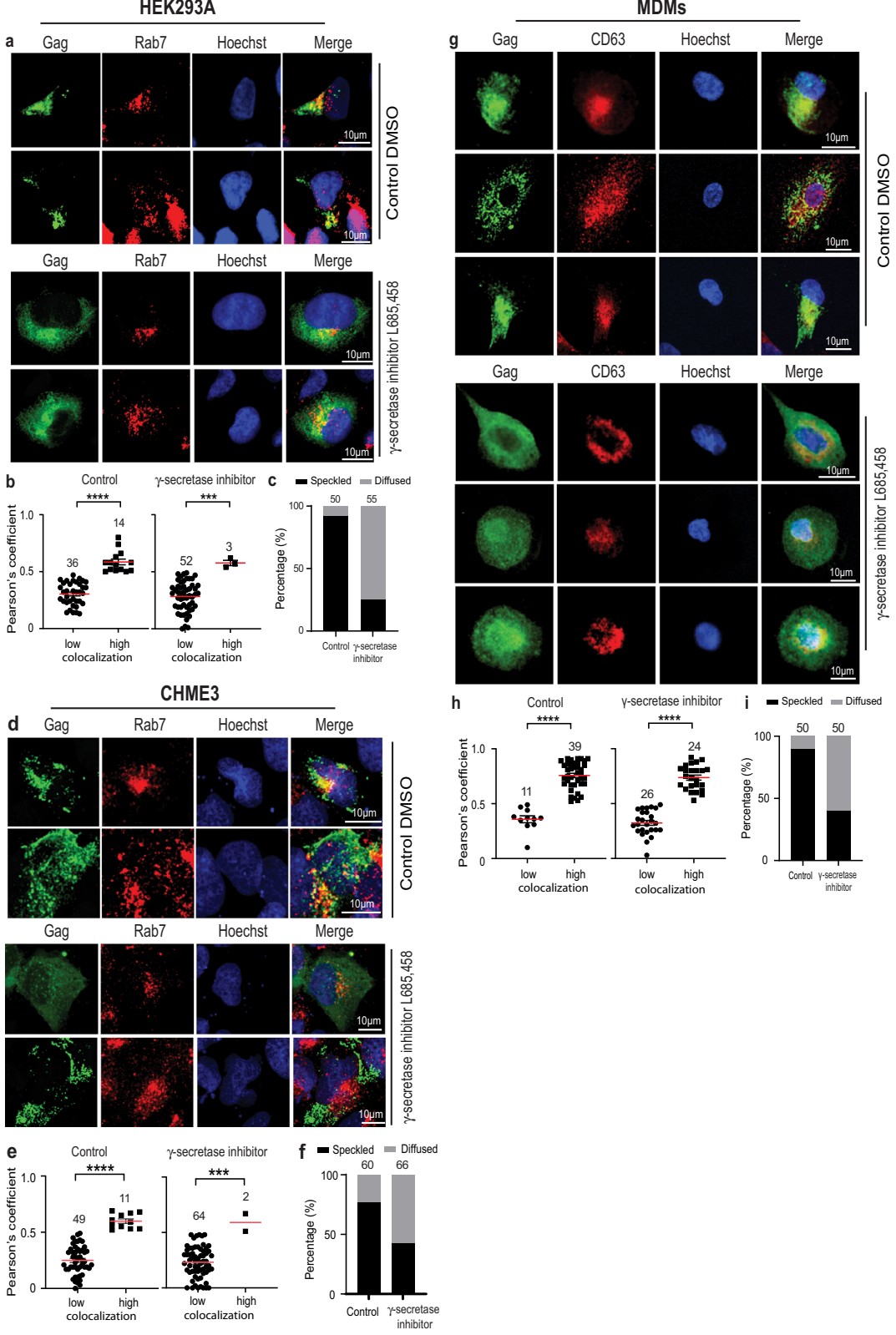

these findings establish that preventing C99 cleavage either through expression of a ubiquitination-defective mutant of C99 or by preventing endogenous APP processing using γ-secretase inhibitors reduces the ability of Gag to access MVB subsets that are known to support HIV-1 replication in both transformed cell lines and biologically relevant immune cell types, such as microglia and primary macrophages.

## Ubiquitination-defective C99 suppresses the production of infectious HIV-1 particles

Defects in Gag localization to CD63-positive MVBs would be predicted to impair HIV-1 replication. Indeed, we have previously shown that γ-secretase inhibitors suppress HIV-1 replication[16]. To determine the potential contributions of amyloidogenic C99 versus non-amyloidogenic C83 cleavage products, we next tested the effects of

**Fig. 6 | Preventing endogenous APP processing limits HIV-1 Gag entry into MVBs.** HEK293A (a–c) or CHME3 (d–f) cells transfected with JR-CSF were treated with DMSO control or γ-secretase inhibitor L685,458 at 4 h post-transfection. 24 h post-transfection, cells were fixed and stained for Gag and Rab7, detecting nuclei using Hoechst. **a, d** Representative images of cells treated with DMSO control or L685,458. **b, e** Quantification of the colocalization of Gag and Rab7 at 24 h using Pearson's Correlation Coefficient; **c, f** Quantification of cells exhibiting typical punctate/speckled versus diffuse Gag distribution patterns. Number of cells analyzed in **b**, **c**, **e** and **f** are indicated. **g–i** MDMs infected with JR-CSF-derived HIV-1 were treated with DMSO control or L685,458 at 24 h post-infection. 5 days post-infection, cells were fixed and stained for Gag and CD63, detecting nuclei using Hoechst. **g** Representative images of cells treated with DMSO control or L-685,458. **h, i** Quantification of the colocalization of Gag and CD63 at 5 days using Pearson's Correlation Coefficient. Data is presented as mean with SEM using unpaired two-tailed *t* test (**b**, γ-secretase inhibitor treatment graph in **e**, **h**, or unpaired two-tailed *t* test with Welch's correction (control graph in **e**). \*\*\**p* < 0.001, \*\*\*\**p* < 0.0001. Source data are provided as a Source Data file.

increasing expression of C83 or C99 on the release of infectious virus from infected cells. Unexpectedly, increasing amounts of exogenous C83, but not GAPDH, increased the levels of extracellular mature virus particles in supernatants from HEK293A cells transfected with either pNL4.3 or JR-CSF, as detected by p24 CA western blot (Fig. 9a, b, respectively). Applying culture supernatants to TZM-bl indicator cells further confirmed that levels of p24 CA in supernatants accurately reflected the levels of extracellular infectious virus particles released from C83 expressing cells in each case (Fig. 9a, b). This may reflect moderate increases in Gag expression that seem to be induced by C83 (Fig. 9a, b) and in line with this, we consistently observed that Gag fluorescence intensity was higher in cells expressing C83 in our earlier imaging approaches. This suggests that perhaps by not interfering with Gag entry into MVBs, C83 actually increases the overall efficiency of infection. By contrast, increasing amounts of exogenous C99 had no effect on p24 CA or infectious virus release (Fig. 9c). This suggests that non-amyloidogenic APP processing is not inhibitory to infection and if anything, may promote infection as discussed later. Moreover, the inability of Flag-C99 to affect infection was unsurprising given that it did not affect Gag localization to MVBs or CD63-positive vesicles (Figs. 3 and 4), in line with ubiquitination of C99 supporting exocytic vesicle trafficking and C99 degradation. However, in line with its effects on Gag localization to MVBs, expression of the C99-7KA ubiquitination-site mutant resulted in a significant decrease in the production of JR-CSF-derived infectious virions in both HEK293A cells (Fig. 9d) and to a significant, albeit lesser extent in CHME3 cells, due to lower transfection efficiency (Fig. 9e). This suggests that the amyloidogenic CTF product, C99 is inhibitory to virus release if it cannot be ubiquitinated by HIV-1.

## Discussion

Beyond providing mechanistic insights into how amyloidogenic processing inhibits HIV-1 replication and its potential as a target to treat HAND, our findings also shed new light on host factors and processes that influence the intracellular vesicle-based replication of HIV-1 in immune cell types such as microglia and macrophages.

Unlike CD4+ T-cells where viral assembly and budding occur primarily at the plasma membrane, HIV-1 also forms large intracellular MVBs that play important roles in the replication and transmission of infectious virus by macrophages and microglia. Studies suggest that when Gag is initially expressed in macrophages it localizes to the plasma membrane and buds as in T-cells, even though significant amounts of Gag are internalized by endocytosis at this time[10,39]. Supporting this idea, disruption of actin-dependent phagocytosis redistributes internalized Gag and virions to the plasma membrane but does not affect virus release[10]. While this was interpreted as meaning MVB's are not a productive route of viral budding, the failure of this redistribution to increase virus release can also be interpreted as meaning budding at the plasma membrane is rapidly saturated in macrophages. Indeed, beyond these early events, as infection progresses the majority of Gag accumulates in modified vesicles that contain markers of LE's or MVB's, in part through continued internalization from the cell surface but largely by direct budding of Gag into MVBs[11,39–43]. Studies from several groups have shown that at these later stages of infection, Gag buds into MVB's using the vesicular VPS

machinery and that this is a major productive pathway in macrophages[11–15,35,36,38–48]. The MVB subsets that support virus replication are characterized by low levels of LAMP1 and tetraspanins such CD53, but high levels of CD63[9,13,14,34–38]. These modified MVBs are spatially and structurally organized by microtubules, polarizing to the periphery to mediate virus spread through cell-cell contacts[42,43,49,50] (reviewed in[51]). Beyond functioning in exocytic release and cell-cell spread to other immune cell types including CD4 + T Cells, these vesicular sites are also thought to function as reservoirs of infection within macrophages[5,51–53]. However, the host factors that influence HIV-1's ability to form and exploit these MVBs as replication sites or "VCCs" remain relatively poorly understood. Notably, capture and internalization of HIV-1 particles by the membrane associated factors Tetherin/BST-2 and Siglec-1/CD169 contributes to the formation of VCCs[50,54–56] but these factors have been suggested to play opposing roles in macrophage to T cell transmission of HIV-1[54,57]. Here, we extend upon our previous discovery that APP is a highly expressed membrane protein in macrophages and microglia that limits HIV-1 infection and provide insights into the underlying mechanisms of both APP-mediated restriction and viral evasion. Specifically, we find that HIV-1 infection does not affect the turnover of the non-amyloidogenic APP product, C83. Furthermore, not only does C83 not inhibit infection but it actually enhances infection. This appears to be connected to an increase in Gag expression in cells expressing C83, which may be due to the fact that C83 exhibits a higher level of plasma membrane localization than the amyloidogenic product, C99 and may therefore not only affect entry into MVBs, but it may also increase internalization of Gag from the plasma membrane into these intracellular replication sites. While determining the underlying mechanism for this enhancement is of future interest, subpopulations of C83 also localized to MVBs yet they are clearly not inhibitory to infection. This suggests that the presence of CTFs at these sites is not sufficient in and of itself to suppress either Gag localization to these vesicles or virus replication. Instead, the negative effects of APP on HIV-1 infection appear to center around its vesicle-based amyloidogenic processing into the larger CTF, C99 that is ubiquitinated to control its intracellular sorting (Fig. 10).

Beyond a role for the γ-secretase component of this processing pathway, which we uncovered previously, here we reveal that HIV-1 stimulates the ubiquitination and degradation of C99. Notably, C99 contains an extended N-terminal region that inserts into membranes and contains an additional lysine residue, which our mutagenesis shows is required for the full inhibitory activity of the C99-7KA mutant. Moreover, our partial mutants with intermediate effects on C99 stability in infected cells align with studies of amyloidogenic APP processing. These studies have shown that ubiquitination plays a key role in engaging the VPS machinery, titling the balance from lysosomal clearance to exocytic release of toxic amyloids[24–29]. Under normal conditions, lysosomal turnover clears amyloids and maintains a healthy state. However, aging, disease or experimental conditions that accelerate APP endocytosis or perturb endosomal sorting to lysosomes results in redirection of APP/C99-containing MVBs to alternative vesicular pathways that increase amyloid secretion, rather than clearance. Sorting of APP to these different sites involves MVBs and specific components of the VPS machinery[25–27,58–60]. Notably, APP and HIV-1 Gag use both overlapping and unique VPS subunits that mediate

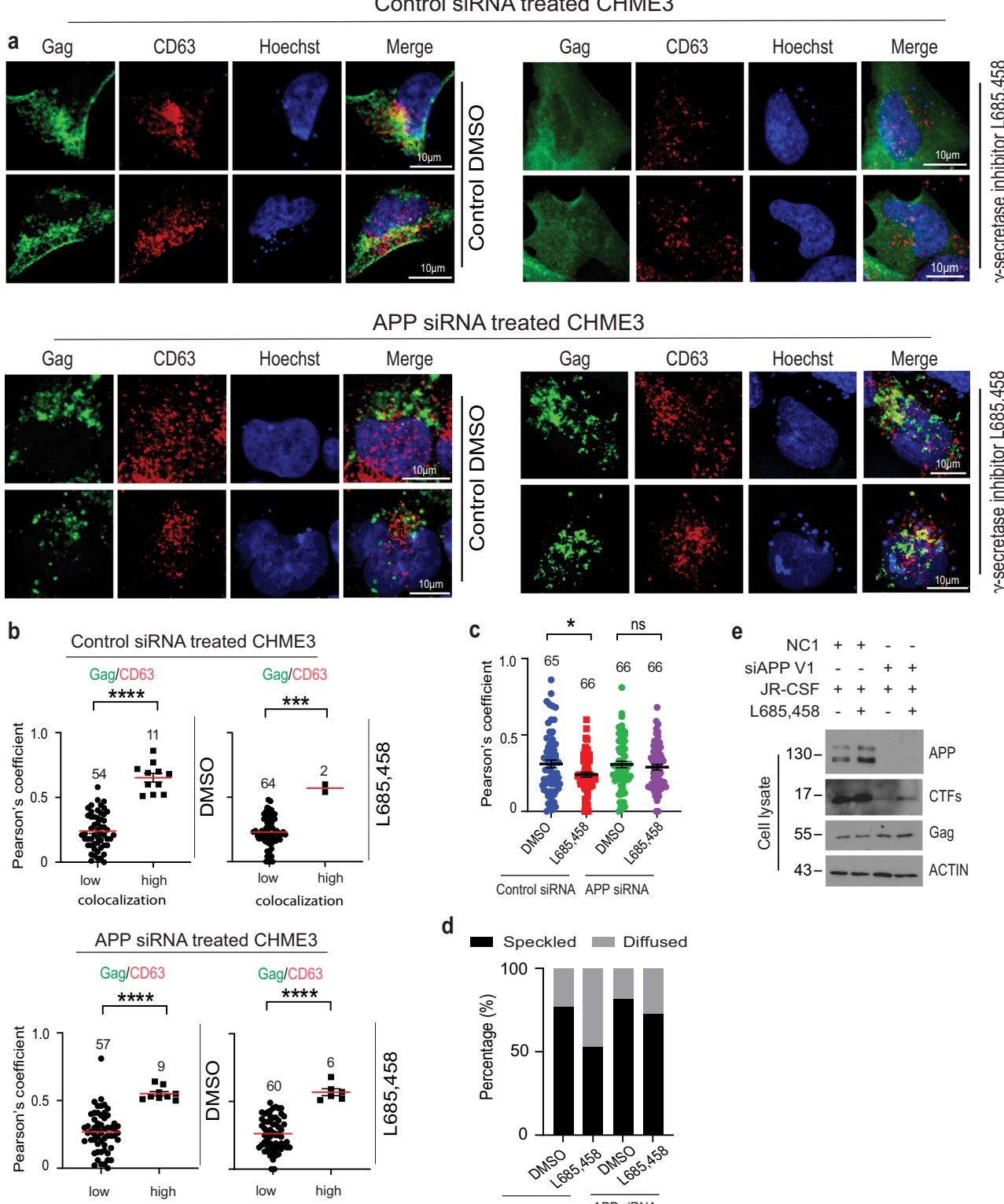

**Fig. 7 | The block to HIV-1 Gag entry into CD63-positive MVBs mediated by γ-secretase inhibition requires APP in CHME3 cells.** CHME3 cells treated with control (**a**) (upper panels) or APP (**a**) (lower panels) siRNAs were transfected with JR-CSF followed by treatment with DMSO control or γ-secretase inhibitor L685,458 4 h post-transfection. 24 h post-transfection, cells were fixed and stained for Gag, CD63, detecting nuclei using Hoechst. **a** Representative images of cells treated with DMSO control (left hand side panels) or L685,458 (right hand side panels). **b**, **c** Quantification of the colocalization of Gag and CD63 under each condition determined by Pearson's Correlation Coefficient; mean with SEM using unpaired

two-tailed *t* test with Welch's correction (APP siRNA and DMSO treated graph in **b**, control siRNA treated graph in **c**) or unpaired two-tailed *t* test for the remaining graphs, *$p < 0.05$, ***$p < 0.001$, ****$p < 0.0001$, ns: not significant. Number of cells analyzed is indicated. Panel (**c**) shows the combined high and low data analysis from **b**. **d** Quantification of cells exhibiting typical punctate/speckled versus diffuse Gag distribution patterns. Number of cells analyzed is indicated. **e** Representative ($n = 3$) WB confirmation of APP and Pr55 Gag levels in samples from **a**. Source data are provided as a Source Data file.

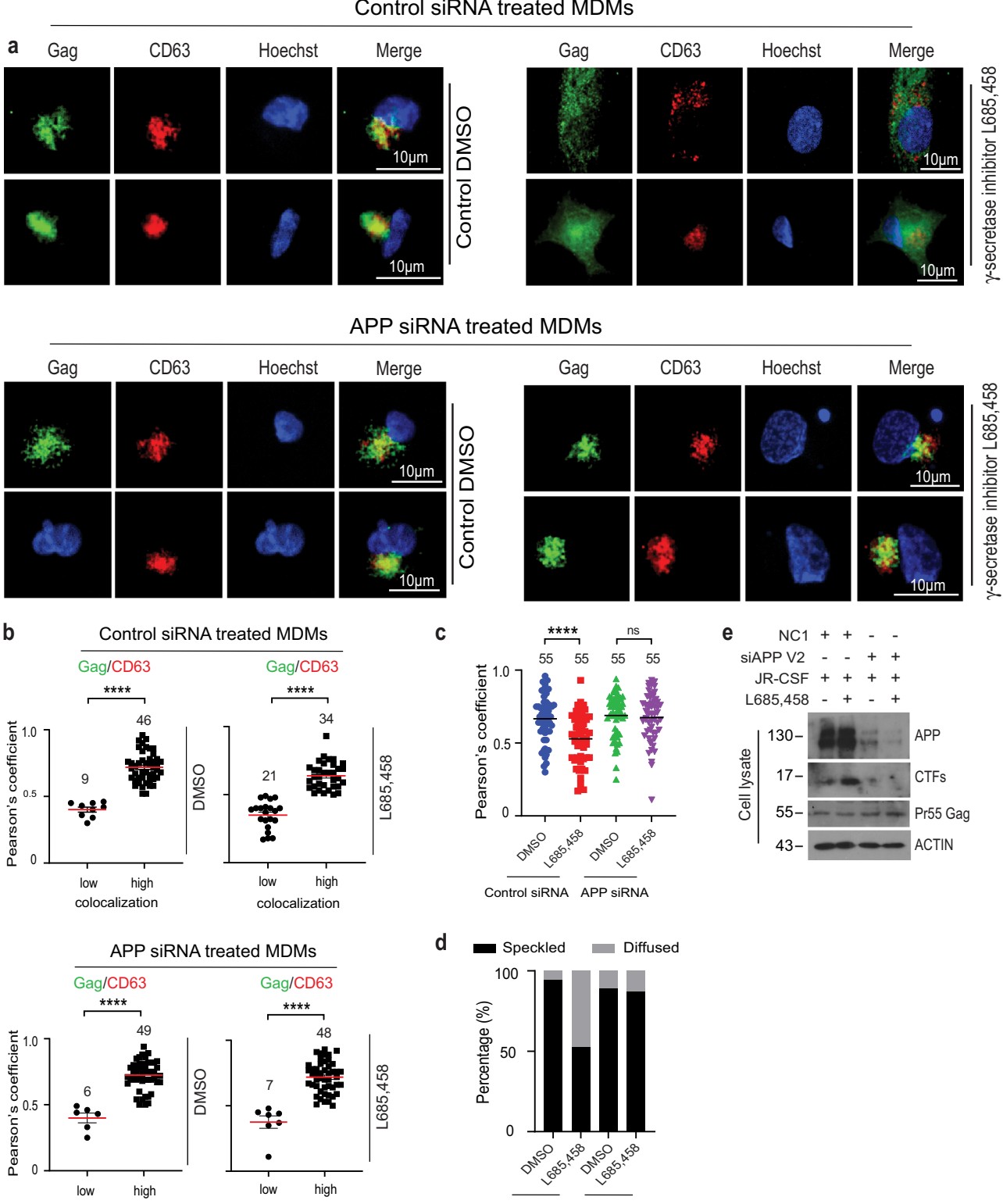

**Fig. 8 | The block to HIV-1 Gag entry into CD63-positive MVBs mediated by γ-secretase inhibition requires APP in primary human macrophages.** MDMs treated with control ((**a**) upper panels) or APP ((**a**) lower panels) siRNAs were infected with JR-CSF-derived HIV-1 followed by treatment with DMSO control or γ-secretase inhibitor L685,458 4 h post-transfection. 24 h post-transfections, cells were fixed and stained for Gag, CD63, detecting nuclei using Hoechst. **a** Representative images of cells treated with DMSO control (left hand side panels) or L685,458 (right hand side panels). **b, c** Quantification of the colocalization of Gag and CD63 under each condition determined by Pearson's Correlation Coefficient;

mean with SEM using unpaired two-tailed *t* test with Welch's correction (control siRNA and DMSO treated graph in **b**) or unpaired two-tailed *t* test for the remaining graphs, ****$p < 0.0001$, ns: not significant. Number of cells analyzed is indicated. Panel (**c**) shows the combined high and low data analysis from **b**. **d** Quantification of cells exhibiting typical punctate/speckled versus diffuse Gag distribution patterns. Number of cells analyzed is indicated. **e** Representative ($n = 3$) WB confirmation of APP and Pr55 Gag levels in samples from **a**. Source data are provided as a Source Data file.

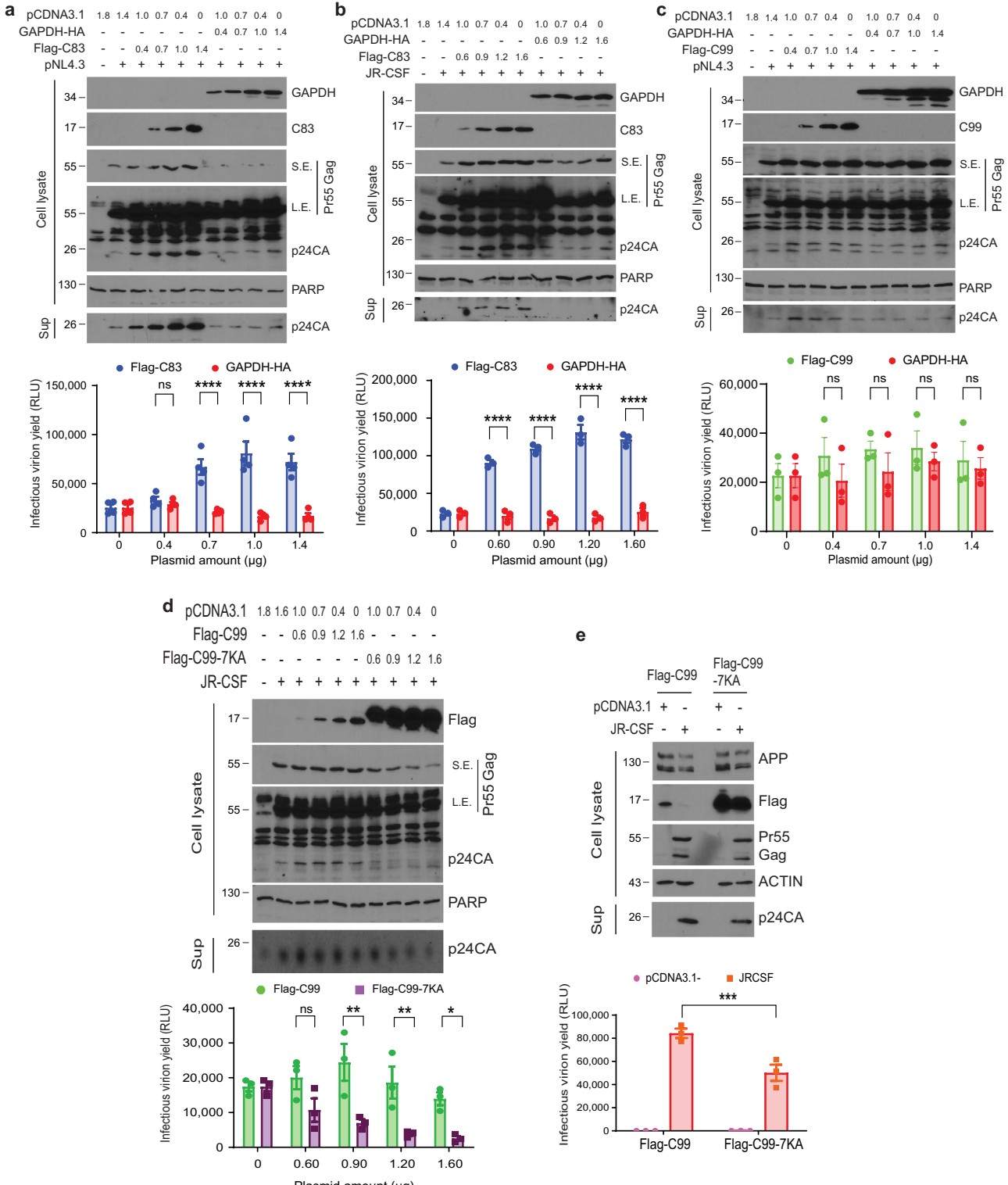

**Fig. 9 | Ubiquitination-defective C99 suppresses the production of infectious HIV-1 particles.** Representative WB analysis showing increasing expression of Flag-C83, but not control GAPDH-HA, increases production of extracellular HIV-1 particles in HEK293A transfected with pNL4.3 (**a**) or JR-CSF (**b**). S.E., short exposure; L.E., long exposure. Note that full-length Pr55 Gag signals become saturated and therefore appear the same in long exposures needed to detect p24 CA. **c** Representative WB analysis showing increasing expression of Flag-C99 or control GAPDH-HA does not affect pNL4.3 virion release in HEK293A cells. **d** Representative WB analysis showing increasing amounts of stabilized Flag-C99-7KA mutant, but

not Flag-C99, results in a dose-dependent inhibition of infectious virus release from HEK293A cells transfected with JR-CSF (note, p24 CA drops to background levels of uninfected cells at the highest C99-7KA concentration). **e** Representative WB analysis showing expression of Flag-C99-7KA decreases extracellular HIV-1 particles in CHME3 cells infected with JR-CSF. **a**–**e** lower panels: Infectious virus in supernatants measured using TZM-bl indicator cells. $n = 4$ (**a**), $n = 3$ (**b**–**e**) shown as mean, SEM; two-way ANOVA Sidak's multiple comparisons test, *$p < 0.05$, **$p < 0.01$, ***$p < 0.001$, ****$p < 0.0001$, ns: not significant. Source data are provided as a Source Data file.

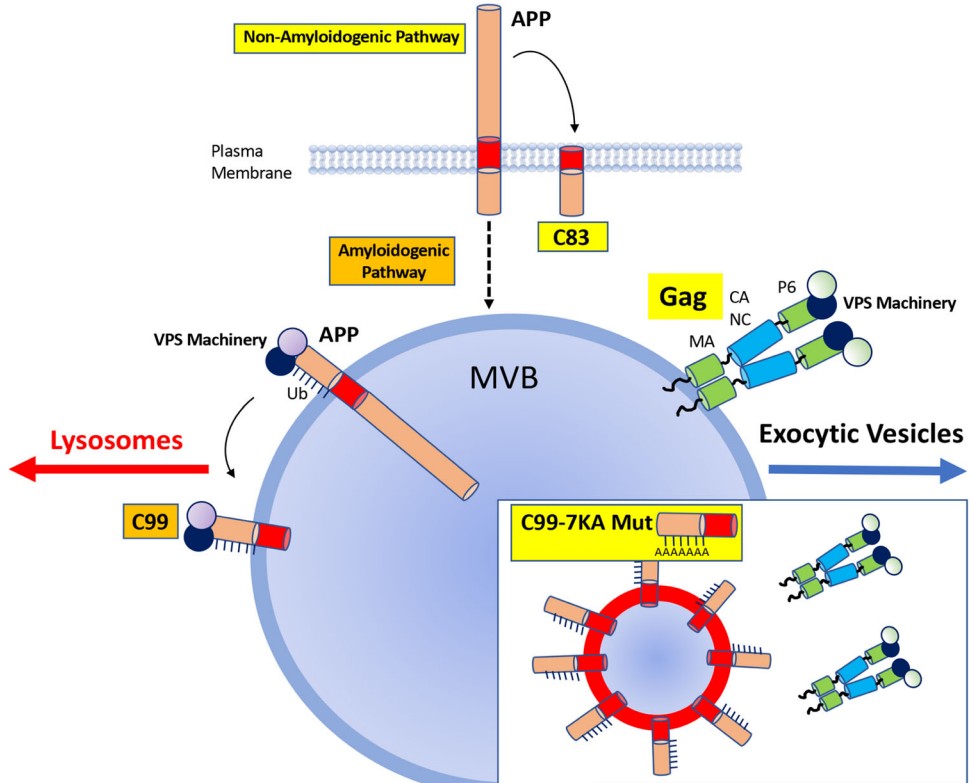

**Fig. 10 | Model for how Gag and C99 compete for control of vesicular sorting pathways.** During amyloidogenic processing of APP, ubiquitination (Ub) of the C-terminal C99 domain regulates invagination into and sorting of MVBs to lysosomes by engaging the VPS machinery. HIV-1 Gag, which comprises of four main structural domains: matrix (MA), capsid (CA), nucleocapsid (NC) and p6, uses a similar strategy to bud into and control sorting of MVBs to exocytic pathways, establishing a competition for control of MVBs using shared and distinct host factors. INSET: The "7-KA" mutant of C99 acts as a dominant-negative that makes MVBs inaccessible to Gag.

their insertion into and control of MVB sorting[4,8–10]. Moreover, we and others have shown that the MA domain of Gag influences membrane targeting[16,23] and we further show here that these MA mutations also impact HIV-1's ability to increase APP processing. Moreover, we reveal that both C99 and Gag target the same CD63-positive MVB subsets. Combined, this suggests that both proteins ultimately intersect at MVBs but with conflicting end-goals of lysosomal clearance for APP under normal cellular homeostasis, versus exocytic release of HIV-1 under pathogenic conditions. To "win out" this battle for control of vesicle sorting, HIV-1 drives exocytic release pathways that result in amyloid release. While C99 ubiquitination could occur passively as a result of simply being drawn into exocytic pathways at a higher rate in infected cells, data using either the C99-7KA mutant or γ-secretase inhibitors suggest that this is also functionally important for the virus to achieve in order to prevent C99 from suppressing its access to MVBs required for replication.

While APP has been implicated in various neuronal and synaptic processes, its biological function beyond producing amyloids remains relatively poorly understood[18]. However, roles in regulating vesicle sorting have been suggested. For example, similar to our findings relating to the specific functionality of C99 in restricting HIV-1 infection, increased expression of APP or C99, but not C83, impairs endocytic sorting and retrograde axonal trafficking, resulting in neurodegeneration[61]. Our data further supports this concept that C99 regulates endocytic sorting and raises the intriguing possibility that one reason that the amyloidogenic pathway is branched between sorting to lysosomes versus exocytic vesicles is to sense and inhibit pathogens such as HIV-1 that exploit MVBs. If C99 wins out, it would either block access to MVBs or direct pathogens to non-productive lysosomes. If C99 loses out, by being directed into exocytic pathways it

may be designed to release amyloids to indicate APP's antiviral activity is being overcome, creating an inflammatory environment or pathogen alert. Indeed, evidence is emerging that APP may function as an antimicrobial protein, with amyloids also proposed to have antimicrobial properties[62–69]. Moreover, APP itself was very recently shown to restrict Zika virus infection in the brain[70]. Finally, as we have shown previously and reinforce here, both the production of amyloids and the antiviral activity of APP can be simultaneously harnessed using clinically approved γ-secretase inhibitors, suggesting that these drugs may be more beneficial in the treatment of HAND than they have proven to date in the treatment of non-HIV-related neurodegenerative conditions such as Alzheimer's Disease.

## Methods

### Ethics Statement
LifeSource Buffy Coat blood obtained from New York blood center was used for isolation of peripheral blood mononuclear cells using Ficoll-Plaque (GE). Ethical approval for the study was obtained from the Institutional Review Board of Northwestern University and all donors provided their written, informed consent.

### Cell cultures
HEK293T cells were obtained from ATCC (cat# CLR-3216). Hela-TZM-bl cells were obtained from NIH AIDS Reagent Program (cat# 8129). HEK293A, CHME3, and CHME3 4 × 4 cells were kindly provided by Jeremy Luban, Marc Tardieu and Olivier Schwarts, respectively as previously described[16]. Primary human monocyte-derived macrophages (MDMs) were derived from CD14+ peripheral blood monocytes by culturing in RPMI medium 1640 (Gibco, 11875-093) with 15% Fetal Bovine Serum (FBS) (Atlanta biologicals, S11150), and 4mM

L-Glutamine (Corning, 25-005-CI). Isolated monocytes were differentiated to MDMs in culture medium supplemented with 50 ng/mL recombinant human M-CSF (R&D, 216-MC/CF) for 10 days. The culture medium was changed every 2–3 days. HEK293A cells were maintained in Dulbecco's Modification of Eagle's Medium (DMEM, Corning, 15-017-CV) supplemented with 10% Nu-serum (Corning, 355500), 100 units/mL penicillin, 100 μg/mL streptomycin (Gibco, 15140-122), and 2mM L-Glutamine. HEK293T cells were cultured in DMEM with 10% FBS, 100 units/mL penicillin, and 100 μg/mL streptomycin for wild type HIV-1 production. The human microglia cell line CHME3, CHME3 4X4, CHME3-derived stable cell lines CHME3-Flag-C99, CHME3-Flag-C99-7KA, and CHME3-Flag-C83 were cultured in DMEM medium with 10% Nu-serum, 100 units/mL penicillin, 100 μg/mL streptomycin, 2mM L-Glutamine, and 1 mM sodium pyruvate (Gibco, 11360-070). Hela-TZM-bl indicator cells were maintained in DMEM medium with 10% FBS, 100 units/mL penicillin, 100 μg/mL streptomycin, and 2mM L-Glutamine. All cells were incubated at 37°C in a humidified environment incubator with 5% $CO_2$.

## Antibodies and inhibitors

Antibodies against APP/CTFs Y188 (ab32136, Abcam, western blotting (WB) 1:1000), APP (LN27, Invitrogen, 130200, immunofluorescene (IF) assay 1:150), ACTIN (A2103, Sigma, WB 1:1000), PARP (9542, Cell Signaling Technology, WB 1:1000), V5 (D3H8Q, 13202, Cell Signaling Technology, WB 1:1000), HIV-1 Pr55/p24/p17(ab63917, Abcam, WB 1:1000, IF assay 1:200) (labeled as Pr55 Gag in the Figures), HIV-1 p24 (ab9071, Abcam, IF assay 1:200), Ubiquitin (P4D1, 3936, Cell Signaling Technology, WB 1:1000), Flag (F7425, Sigma, WB 1:1000), Flag (L5, NBP1-06712, Novus, IF assay 1:100), Flag (M2, Sigma, see immunoprecipitation assay section), HA (H3663, Sigma, WB 1:1000), Rab7 (D95F2, 9367, Cell Signaling Technology, IF assay 1:200), EEA1 (C45B10, 3288, Cell Signaling Technology, IF assay 1:200), CD63 (H5C6, DSHB, IF assay 1:100), LAMP1 (1D4B, sc-19992, Santa Cruz, IF assay 1:100) and UBE1 (15912-1-AP, Proteintech, WB 1:1000) were used according to manufacturer's instructions. Secondary HRP linked antibodies against rabbit IgG (NA934) and mouse IgG (NA931) were obtained from GE Healthcare UK and used at 1:10000 dilution. Alexa fluorescence-conjugated secondary antibodies against mouse IgG 647 (Invitrogen, A31571), mouse IgG 488 (Invitrogen, A21202), mouse IgG 555 (Invitrogen, A31570), rabbit IgG 647 (Invitrogen, A31573), rabbit IgG 488 (Invitrogen, A21206), rabbit IgG 555 (Invitrogen, A31572), rat IgG 647 (Jackson ImmunoResearch, 712-605-150) were used at 1:400 dilution. Bafilomycin A1 (BafA1, B1793) and MG132 (474790) were purchased from Sigma Aldrich. TAK-243 (HY-100487) was purchased from MedchemExpress (MCE). L685,458 (2627) was purchased from Tocris. Cycloheximide (CHX, 357420010) was purchased from Acros Organics.

## Western blotting (WB)

Cells were lysed in Laemmli buffer and boiled for 3 minutes (min) prior to loading. Proteins were fractionated by SDS-PAGE and then transferred to nitrocellulose (NC) membrane or polyvinylidene difluoride (PVDF) at 70 V for 1 hour (h). The membrane was blocked in blocking buffer (TBST containing 3% non-fat milk) rocking at room temperature for 1 h. After washing 3 times in TBST, the proteins of interest were probed with primary antibodies according to manufacturer's instructions rocking at 4 °C overnight. Appropriate secondary HRP linked antibodies were incubated rocking at room temperature for 1 h, followed by 3 times wash with TBST. Membranes were incubated with Pierce ECL substrate (Thermo Fisher, 32106) and then exposed to X-ray film using Konica SRX-101 medical film processor. Uncropped and unprocessed scans of all blots are supplied in the Source Data file. Images were acquired using HP ENVY 5530 scanner at 300dpi and quantified using Image J software version 2.3.0. Protein band densitometry was normalized to the respective internal control (ACTIN or

PARP or eIF4E). GraphPad Prism version 9.4.1 was used to create graphs and perform statistical analysis.

## Inhibitor treatment

Cells were treated with BafA1 at 40 nM or MG132 at 20 μM 8 h prior to harvesting the samples. For TAK-243 treatment experiment, HEK293A cells were seeded in 6-well plate with $3 \times 10^5$ cells/well. The next day, a combination of 0.2 μg JR-CSF or pCDNA3.1 with 0.5 μg Flag-C99 or C83-V5 or GAPDH-HA was incubated with 2.8 μL TurboFect for 20 min at room temperature. TAK-243 was applied to the cell 24 h post-transfection at the concentration of 100 nM and maintained for 24 h through the entire experiment. HEK293A or CHME3 cells were treated with L685,458 at 1 μM 4 h post-transfection and maintained until cells were harvested. MDMs were treated with L685,458 at 1 μM one-day post infection and maintained throughout the experiment. Cycloheximide was applied to cells at the concentration of 100 μg/mL 24 h before collecting cell samples for WB analysis.

## Plasmids

HIV-1 NL4-3 infectious clone pNL4.3 (Cat # ARP-114) and JR-CSF infectious clone pYK-JRCSF (Cat # ARP-2708) were obtained from NIH AIDS Reagent Program. Lentiviral expression vector pCDH-CMV-MCS-EF1a-Puro (System Biosciences, Cat # CD510B-1), and the simian immunodeficiency virus (SIV) plasmid pSIV3+ were kindly provided by Dr. Judd Hultquist and Dr. Thomas Hope (Northwestern University, IL), respectively. The HIV-1 pNL4.3 based clones with single mutation at 85YG or 87VE within MA domain were kindly provided by Dr. Eric Freed (National Cancer Institute, MD). The C-terminally HA-tagged human GAPDH (GAPDH-HA)[16], the C-terminally HA-tagged Rev-independent HIV-1 Gag (Gag-HA)[16] and the Nicastrin with C-terminal HA-tag (NCT-HA) (a kind gift from Dr. Huaxi Xu at Sanford Burnham Prebys Medical Discovery Institute, CA) were used as template in PCR amplification. Amplified GAPDH-HA and Gag-HA were then subcloned into the pQCXIN retroviral vector (Clontech) at NotI and EcoRI restriction enzyme sites. NCT-HA was subcloned into the pQCXIN retroviral vector at NotI and PacI sites. APP C-terminal fragment C99- and C83 (Flag-C99 and Flag-C83, respectively) were amplified by PCR using previously described pCAGOSF-APP₇₇₀[16] as template. APP C-terminal fragment C83 with V5-tag on the C-terminus (C83-V5) was amplified by PCR using pCAGOSF-APP₇₇₀ as template. Amplified Flag-C99, Flag-C83, and C83-V5 were subcloned into pQCXIP vector at NotI and PacI sites. The C99 site mutants K687A, K699A, K687,699A, K724,725,726 A, K751A, K763A, and 7KA were generated by PCR-site directed mutagenesis using pQCXIP-Flag-C99 as template. The lentiviral expression vector pCDH-Flag-C99 and pCDH-Flag-C99-7KA both with Flag-tag on the N-terminus were generated by PCR amplification using pQCXIP-Flag-C99 and pQCXIP-Flag-C99-7KA as template, respectively, followed by subcloning into pCDH-CMV-MCS-EF1-Puro vector at XbaI and NotI restriction enzyme sites. Primers used are listed the Supplementary table 1. Sequence integrity of each construct was validated by sequencing.

## HIV-1 stock preparations and infections

To generate wild type virus, pNL4.3 or pYK-JRCSF were transfected into HEK293T cells using polyethylenimine (PEI, Polysciences, Inc. 1 mg/mL)[71]. The culture supernatants were filtered through a 0.45 μm filter, and viral particles were concentrated by passing through a 20% sucrose cushion by ultracentrifugation at $32,000 \times g$ for 2 h at 4 °C. The viral pellet was re-suspended with PBS and stored at −80 °C. For infection, $2 \times 10^4$ CHME4×4 cells per well of 24-well plates were infected with pNL4.3 or pYK-JRCSF-derived HIV-1 containing polybrene (sc-134220, Santa Cruz) at the concentration of 10 μg/mL, followed by replacement of culture medium one day after infection. At the indicated time, cells were harvested for WB analysis. For MDM infection

with wild-type HIV-1, $1.5 \times 10^5$ cells were seeded in the 24-well plate, or $5 \times 10^5$ cells in a 6-well plate over coverslips, a day before infection. The following day, cells were infected with JRCSF-derived HIV-1 with polybrene at the concentration of 10 μg/mL overnight, followed by replacement of culture medium one day after infection. At the indicated time, cells were lysed for WB analysis, and coverslips were fixed for immunofluorescence analysis.

## Plasmid transient transfections

HEK293A cells were transfected with TurboFect reagent (Thermo Scientific, R0531) at a ratio of 1 μg DNA: 4uL of TurboFect reagent. CHME3 and HEK293T cells were transfected at the ratio of 1 μg DNA with 2.25 μL PEI as previously described[71].

## Generation of CHME3 stable cell pools

The murine leukemia virus (MLV)-based vector pQCXIP, encoding Flag-C99 or Flag-C99-7KA or Flag-C83 described in the plasmid section above were used to generate pQCXIP-based retroviruses followed by infection and generation of CHME3 cell pools stably expressing Flag-C99 or Flag-C99-7KA or Flag-C83 as previously described[71]. Specifically, 100% confluent HEK293T cells in a 10 cm dish were split at the ratio of 1:4. The following day, a combination of 4.3 μg of transfer plasmid, 2.85 μg of MLV Gag-Pol expressing vector pCMV-intron, and 2.85 μg of vesicular stomatitis virus G envelope protein expression plasmid pVSV-G was mixed with 22.5 μL PEI in 1 mL Opti-MEM, followed by 20 min incubation at room temperature. The DNA mixture was then gently added onto the cells dropwise. The next day, culture medium was replaced with fresh media and supernatants were filtered through a 0.45 μm filter, aliquoted and frozen at −80 °C at 48 h post transfection. CHME3 cells seeded in a 6-well plate at $5 \times 10^4$ cells/well were infected with the filtered supernatants containing Flag-C99 or Flag-C99-7KA retroviruses in the presence of polybrene (sc-134220, Santa Cruz) at 10 μg/mL, followed by replacement of culture medium to fresh medium 24 h later. Infected cells were selected in culture medium containing 2 μg/mL puromycin (P8833, Sigma) until all uninfected control cells were dead. WB analysis was used to determine the expression of proteins using adequate antibodies.

## Lentiviral transduction of MDMs

The lentiviral expression vector pCDH, encoding Flag-C99 or Flag-C99-7KA described in the plasmid section, was used to generate pseudotyped lentiviruses followed by transduction of MDMs. Briefly, a full confluent 10 cm dish of HEK293T cells were split at the ratio of 1:4. The following day, each dish of cells was transfected with a combination of 2.5 μg expression vector, 1.66 μg lentiviral packaging plasmid (p8.91), and 0.83 μg pVSV-G using 22.5 μL PEI. Viral supernatants were filtered through 0.45 μm filters, aliquoted and frozen at −80°C at 48 h post transfection. VSV-G-pseudotyped SIV containing Vpx was used to improve lentivirus infection efficiency and was produced by transfecting HEK293T cells with 7.5 μg pSIV3+ and 2.5ug pVSV-G using 22.5 μL PEI. The supernatants collected at 48 h and 72 h post transfection were filtered through 0.45 μm filters, combined, and frozen at −80 °C. MDM cells were seeded into 6-well plates over coverslips at a density of 3-5 x $10^5$ cells per well 24 h prior to transduction. Cells were pretreated with viral like particle SIV3+ with 10 μg/mL polybrene for 5 h, followed by infection with pseudotyped lentivirus with 10ug/mL polybrene overnight. Cell culture media was changed the day after transduction, and cells were subjected to the indicated treatment.

## Immunofluorescence assay

HEK293A, CHME3, CHME3-derived stable cells or MDMs were seeded on coverslips at $3 \times 10^5$ cells per well in a 6-well plate. At indicated time, cells were washed with PBS once and then fixed in 4% paraformaldehyde for 15 min. Fixed cells were washed with PBS three times and then permeabilized in 0.1% Triton X-100 PBS for 30 min at room

temperature. After 3 times wash with PBS, cells were blocked in PBS containing 10% donkey negative serum (DNS) and 0.25% saponin for 40 min and then incubated with primary antibodies in PBS with 10% DNS and 0.025% saponin overnight at 4 °C. The following day, cells were washed using washing buffer (PBS containing 0.025% saponin) for 3 times and thereafter incubated with appropriate Alexa fluorescence-conjugated secondary antibodies at 1:400 ratio in PBS with 10% DNS and 0.025% saponin for 1 h at room temperature. Cell nuclei was stained with Hoechst 33342 at 1:3000 in washing buffer for 10 min, followed by 3 times wash with washing buffer. Cells were imaged using Leica DMI 6000B microscope. Images were visualized and analyzed using the Metamorph imaging software version 7.10.5.476. Image J software version 2.3.0 was used to quantify colocalization.

## Immunoprecipitation (IP) assay

For IP assay, $2 \times 10^6$ HEK293A cells were plated in a 10 cm dish a day prior to transfection. A combination of 2 μg pCDNA3.1 or JR-CSF with 5 μg pQCXIP-Flag or pQCXIP-Flag-C99 or pQCXIP-Flag-C99-7KA as indicated was transfected using TurboFect. 48 h post-transfection, cells were rinsed with 10 mL prechilled PBS once and lysed by adding 750 μL ice-cold CHAPS buffer (50 mM HEPES buffer PH7.4, 2 mM EDTA, 150 mM NaCl, 2 mM $Na_3VO_4$, 0.5% CHAPS, 1.5 mM $MgCl_2$). After rocking at 4 °C for 40 min, samples were spun down at 13,000 rpm 4 °C for 10 min to remove the debris. Supernatants were transferred to fresh tubes, followed by adding 25 μL slurry of 10% G-sepharose (GE Healthcare, 17-0618-01) to preclear for 40 min at 4 °C, and beads were spun out. 27 μL of supernatant was collected and combined with 33 μL of Laemmli buffer as input. The remaining supernatant was incubated with 4 μL Flag M2 antibody and 40 μL of 50% slurry equilibrated beads for 3 h rocking at 4 °C. Beads were washed with 700 μL CHAPS buffer 3 times for 5 min each rocking at 4 °C. Briefly centrifuged beads were resuspended with 50 μL of Laemmli buffer (referred to as bound) and then boiled for 3 min for WB analysis.

## Infectious virion yield assays

Infectious virus yields were determined using Hela-TZM-bl indicator cells followed by measurements of beta-galactosidase activity as previously described[16]. Briefly, Hela-TZM-bl cells were seeded in a 96-well plate at $1.5 \times 10^4$ cells/well. The next day, the cells were inoculated with 100 μL of filtered serially diluted supernatants. 48 h post infection, cells were washed with ice-cold PBS twice and lysed by adding 30 μL of lysis solution to each well provided by Galacto-Star system (Applied Biosystems). After 10 min incubation, cell lysates were transferred to a microfuge tube, followed by centrifuging for 2 min to pellet cell debris. 2 μL of supernatant was transferred to a luminescence microplate and incubated with 100 μL of reaction buffer with 2% Galacton-star substrate for 1 h at room temperature. The signal was measured by a luminometer for 1 s per well.

## RNA interference

Cells were seeded on coated coverslips in 6-well plates in culture medium without antibiotic one day prior to transfection. Lipofectamine RNAiMax transfection reagent (Invitrogen, 13778-150) was used to carry siRNA duplexes into cells. To prepare the mixture A, 3 μL of each siRNA was combined with 100 μL of Opti-MEM. For making mixture B, 4 μL of RNAiMax reagent was combined with 50 μL of Opti-MEM. After 5 min incubation, mixture A was mixed with B by pipetting up and down gently, followed by incubating for 20 min at room temperature. During incubation, cell culture media was replaced by 1 mL of pre-warmed Opti-MEM. Thereafter, the RNAi mixture was added onto the cells dropwise, followed by the addition of 1 mL of fresh culture medium without antibiotic 4 h later. The following day, culture medium was replaced by 2 mL fresh medium. Depletion of protein of interest was confirmed using WB or immunofluorescence analysis.

Cells were used in different assays 48 to 72 h post-transfection. siRNA information can be found in the Supplementary table 2.

### Statistical analysis

Adobe Photoshop 2022 version 23.3.0.394 and illustrator 2022 version 26.2.1 were used to create figures. GraphPad Prism 9.4.1 was used to create graphs and perform statistical analysis. Data were analyzed by one sample $t$ and Wilcoxon test for one group or some data with identical control value, two-tailed student's $t$ test for two-group comparisons and one-way ANOVA or two-way ANOVA followed by post hoc test for 3 or more groups by GraphPad Prism software version 9.4.1. All data were presented as mean ± SEM. The $p$ that was less than 0.05 was considered significant. $*p < 0.05$, $**p < 0.01$, $***p < 0.001$, $****p < 0.0001$, ns: no significant.

### Reporting summary

Further information on research design is available in the Nature Portfolio Reporting Summary linked to this article.

## Data availability

The authors declare that the data supporting the findings of this study are available within the article and its Supplementary Information files. Source data are provided with this paper.

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

## Acknowledgements

We thank Dr. Huaxi Xu at Sanford Burnham Prebys Medical Discovery Institute for the HA-tagged nicastrin construct, Dr. Judd Hultquist and Dr. Thomas Hope at Northwestern University for the lentiviral transduction system and Dr. Eric Freed at Center for Cancer Research, National Cancer Institute for MA mutant constructs 85YG and 87VE. The following reagents were obtained through the NIH AIDS Reagent Program, Division of AIDS, NIAID, NIH: pNL4.3 from Dr. Malcom Martin, JR-CSF from Dr. Irvin S.Y. Chen and Dr. Yoshio Koyanagi, and TZM-bl from Dr. John C. Kappes, Dr. Xiaoyun Wu and Tranzyme Inc. This work was supported by NIH grants R01 NS099064 and R01 NS131094 to M.H.N.

## Author contributions

F.G. and M.H.N. designed research; F.G. and M.B. performed research and analysed data; and M.H.N. wrote the paper.

## Competing interests

The authors declare no conflict of interest.
