## [Peer Review File · Nature Communications]

HIV-1 promotes ubiquitination of the amyloidogenic C-terminal fragment of APP to support viral replicationREVIEWER COMMENTS

Reviewer #1 (Remarks to the Author):

Current manuscript Gu et al from the Naghavi lab is built upon a successful publication in Nature Communication back in 2017 (Chai et al 2017 DOI: 10.1038/s41467-017-01795-8). This earlier work has mapped the amyloid related biology onto the Gag amino acids somewhere between aa77-111 within the matrix domain. A focus of this early work was on the impacts of membrane protein targeting resulting from the interplay between HIV Gag and amyloid. Previous work takes advantage of transfection system in cell lines. Current manuscript continuously relies on transfection system with cell lines, pharmacological agents, immuno-blots, fluorescent imaging, and pull down. Given the success of this team on their prior 2017 paper (plus the knowledge gained then), this assessor was hoping (perhaps unfairly) that some of the more relevant systems, approaches, or constructs would have been used by authors on their current manuscript / investigation.

This is an important study toward the relationship between HIV and amyloidogenic pathway. Current manuscript has been written in a way that 'somewhat assumed' readers are cell biologists who already have acquired expertise in HIV biology, amyloid biology, or both. I would encourage authors to bring presentation to the levels of general audience to enhance its impact. For example, Fig1A from the Naghavi lab 2017 paper (Chai et al 2017 DOI: 10.1038/s41467-017-01795-8) nicely capture the amyloid processing biology and set the tone for their 2017 paper back then. Something like such graphic representation in some of the figures would help message delivery.

Current story feels somewhat incomplete for this assessor. Whether it is the need to use more relevant model system (infection [or even single round transduction]) into relevant primary cells, biophysical studies to reveal mechanisms better, and / or more appropriate mutagenesis analyses; are needed (in the view of this assessor) to discern the HIV Gag and amyloid relationships. Selected additional approaches (such as iPC brain organoid, and/or proteomics analyses on pull-down samples) may also enrich the current story.

I really would like to emphasise that the story being presented is sound, and likely to be a great one. I just feel that perhaps some of the 'I's and 'T's need to be dotted and crossed with more rigorous data and interpretations.

- Intracellular vs extracellular budding of HIV in macrophages has been a major debate. In these prior works, multiple studies use primary macrophages. One clear message would be that fluorescent imaging has its limitation on discerning intracellular budding (such those in VCC) vs extracellular release. With the current data presented by authors, perhaps it is more appropriate to call them C63 association but stay away from VCC targeting. In particular, both 293 based or CHEM3 based systems have limitations.
- Works of 2017 (Chai et al 2017 DOI: 10.1038/s41467-017-01795-8) from the Naghavi lab have mapped the involvement of MA aa aa77-111 within the HIV Gag in amyloid biology. Interestingly, early work from the Freed lab back in 2000, Ono et al (<https://doi.org/10.1128/JVI.74.6.2855-2866.2000>) reported that mutations between residues 84 and 88 of the matrix (MA) domain of HIV-1 Gag cause a retargeting of virus particle formation to an intracellular site. The overlapping of Gag regions between these biology perhaps should be considered / discussed by authors more extensively, rather than a general single group statement in line 330 page 16 that is amongst 20 references.
- Primary blood- (or monocytes-) derived macrophages and microglia (brain macrophages) are going to be different from 293 and CHEM3 cells in many major ways. In practice, blood cells derived macrophages are much easier to study than isolating resident microglia cell lines (in my view). As abstract described 'HIV infection in macrophages and microglia (3rd sentence). I was left with impression (and expectation that authors will provide data on 'infection of macrophages with brain isolates, such and JR-CSF'. However, there is no data on HIV INFECTION of blood macrophages as I can see. Perhaps, validation data from more NATURAL HIV INFECTION of blood macrophages would be a good way to overcome inherited limitation on current transfection system. Should authors not choose to incorporate macrophage INFECTION data, abstract / summary description needs to be much more 'controlled' to reflect the biological data being

presented – for example microglia cell line would be appropriate.

- Data in Figure 1 is OK. However, is transfection the 'right' system to ask question on biology of HIV in CNS resident cells? I seem to recall expression of HIV gag in microglia cells is not high to begin with Thereby, transfection system will largely skew the effect via over-expression.
- In line 129, authors stated 'infection' of CHME3 4x4. Based on materials and methods, the work described is transfection but not infection. Infection is misleading. Transfection is generally over-expressing plasmid encoded genes.
- Lines 139-141. This reviewer has reservation on the claim that HIV Gag accelerate pathway of APP cleavage. Based on current data, it appears to be a conjecture. Pulse-Chase experiments can however demonstrate this claim nicely and directly.
- Given it is difficult to demonstrate HIV expression in microglia cells in vivo, it is ok to use tissue culture system. Having said that, a more appropriate model (or validation) might be using lentiviral transduction system to control the (titre) gene delivery at 1 copy of gene to be delivered per cell
- Legend in Fig 1 said 'C99'. Was there specific information on Ab to indicate it was not C83? A descriptive sentence on 6E10 would be helpful for non-APP readers.
- Legend in Fig1e-f used the word 'uninfected cells', it will be important to use the 'word' un-transfected' instead. While current description is 'correct' but can be a bit misleading given the data in Fig1 is transfection system
- Not exactly sure the value of the 'time course' in figure 1. First, there is no obvious differences with data between time points. Second, these are transfection data, the system is already overwhelmed with 'over-expression' from plasmid in comparison with natural infection processes. Perhaps remove time point data in Fig 1.
- Fig2a-b onFlag-C99 vs Flag-C83 are nice beginning – I would thought that authors could have taken advantage of large body of accumulated Gag point mutants (with known biology) in public domain to evaluate how these HIV Gag would have impacted on amyloid processing (non-amyloidgenic- vs non-amyloidgenic-pathways). More specifically, rather than introducing FlagC-99 and Flag-C83 separately, would it not be more informative to introduce N-terminal tagged FLAG-APP and C-terminal tagged APP-HA in parallel experiments to perform pulse chase in the presence vs absence of HIV Gag – would it not be more informative to show the dynamic of the APP processing starting from full-length APP?
- Pharmacological agents (to me) are helpful with mechanism dissection, but they have issues (in my view) of pleiotropic effects – I consider supportive information; hence I would prefer to see additional experimental approaches.
- Mutations in Fig 2f-g can only illustrate ala mutations led to reduction of protein levels, but NOT NECESSARY via ubiquitination.
- Also, there is an alignment issues for western data in Fig 2f and the densitometry data in Fig 2g. There are 18 samples in Fig 2f, but only 16 densitometry analyses in Fig2g. I suspect these are related to the mock b-CTF transfection control lanes (in the first) two. Readers however should not have to guess, correction on these would be important.
- With Fig2h that authors already have the immunoblots on ubiquitination, I do not agree with authors' comments on Flag-7KA does not have ubiquitination. Different levels of ubiquitination between FLAG-C99 and FLAG-7KA sure, but no ubiquitination as stated in legend would be 'incorrect' for me. Given these samples are ready, perhaps it would be wise to cut out these bands from PAGE to perform a target proteomics on C99 WT and mutants to get the profile. Commercially available proteomic quantification kits may also be used to measure potential differences on levels on post-translational modifications.
- Associations amongst Gag, C99, Rab7, and CD63 in Figs 3-4 are good data. Claims on 'Preventing C99 ubiquitination' would be overstated in my view. C-99-7KA has its limitation. One will need quantitative proteomics first to make such statement. 293 and CHME3 cells have their limitations – primary cell system would have been better.
- Just want to be clear – I am comfortable with the reported effects authors made with C-99-7KA. I am just less confident that C-99-7KA is clean enough to be used as ubiquitination negative situation.
- Also, if the argument is that introducing these protein fragments to alter neuropathology, would it not be more critical to evaluate the impacts of processing of endogenous amyloid precursor proteins?
- siRNA study is nice – I would like to see that in relevant system though – primary macrophages for example.

- Data in Fig5 on gamma-secretase inhibitors are nice to show significant difference at the imaging level – encouraging. However, given the limitation of ‘over-expression system’ in ‘less relevant cell line systems’, this assessor will have to reserve judgement. It would have been nice transduction system (at low MOI of 1 to reflect most infected macrophages are likely to have one inserted copy of HIV) into primary macrophages (better yet iPC microglia cells). As it is ‘presumed’ that HIV infected microglial cells are derived from transmigration of infected monocytes across the BBB, the ‘simpler version’ (than real microglial cells) of blood monocytes derived macrophages should be sufficient in demonstrating this claim by the authors in a setting that is closer aligned to the situation in vivo.

Reviewer #2 (Remarks to the Author):

Amyloid precursor protein (APP) has been recently identified as an inhibitor of HIV-1 infection in macrophages and microglia but the underlying mechanism remains unknown. Here the authors reveal that entry of HIV-1 Gag into VCCs is blocked by the C-terminal “C99” product that is generated by amyloidogenic processing of APP, but not by the non-amyloidogenic product “C83”. To counter this, Gag promotes multi-site ubiquitination of C99 which controls both exocytic sorting of MVBs and further processing of C99 into toxic amyloids. Processing of C99, entry of Gag into MVBs and release of infectious virus could all be suppressed by expressing ubiquitination-defective C99 or by gamma-secretase inhibitors. The experiments are well executed, provide clear results and display appropriate controls. The findings are interesting as they suggest a mechanism that might suppress the intracellular replication of HIV-1 in brain-resident microglia. However, given that the relevance of the findings is connected with the possibility of treating HIV-associated dementia (HAND), the manuscript does not provide evidence that this strategy might be really beneficial against HAND, as no animal model of HAND is tested in the experimental design of the manuscript. I would consider the whole manuscript as a preliminary study in support of the possibility of using the strategy in vertebrate systems.

Reviewer #3 (Remarks to the Author):

This manuscript extends Dr. Moghavi’s earlier work on the interactions between HIV-1 Gag protein and amyloid precursor protein (and its fragments) and the implications of these interactions for HIV-induced neurological disease. The manuscript presents extensive data supporting the hypothesis that the APP fragment C99 interferes with the trafficking of HIV Gag into multivesicular bodies, and that Gag in turn drives the ubiquitination and resulting degradation of C99.

In general I found the manuscript clear and convincing, and a useful addition to the literature on the important problem of how HIV infection induces neurological disease. There are a few minor issues that should be clarified.

1. I don’t see any mention of the nature or source of the 6E10 monoclonal antibody.
2. The Westerns in Fig. 7 have sections labeled “Pr55 Gag S.E.” and “L.E.” After much head-scratching I decided these must be Short Exposure and Long Exposure. But that doesn’t seem right either, as the bands in left half of panel a S.E. are much darker than those in the right half of panel a S.E., but they seem equivalent in the L.E. set. Some explanation would certainly help.
3. In the description of the production of pQCXIP vector particles, the list of plasmids used includes “pCMV-intron” (line 482). What intron and for what purpose? Is this an insert accompanying some coding region in a plasmid?

We thank the Reviewers for their kind and supportive comments, as well as their insightful and constructive suggestions that have greatly improved this manuscript. We hope that our revisions and responses to the Reviewer's comments, below, address their concerns satisfactorily.

REVIEWER COMMENTS

Reviewer #1 (Remarks to the Author):

Current manuscript Gu et al from the Naghavi lab is built upon a successful publication in Nature Communication back in 2017 (Chai et al 2017 DOI: 10.1038/s41467-017-01795-8). This earlier work has mapped the amyloid related biology onto the Gag amino acids somewhere between aa77-111 within the matrix domain. A focus of this early work was on the impacts of membrane protein targeting resulting from the interplay between HIV Gag and amyloid. Previous work takes advantage of transfection system in cell lines. Current manuscript continuously relies on transfection system with cell lines, pharmacological agents, immuno-blots, fluorescent imaging, and pull down. Given the success of this team on their prior 2017 paper (plus the knowledge gained then), this assessor was hoping (perhaps unfairly) that some of the more relevant systems, approaches, or constructs would have been used by authors on their current manuscript / investigation.

We appreciate the Reviewer's comments and while we feel that many of the approaches listed above are appropriate to address the questions in hand, we agree with the overall comment regarding the importance of extending our studies to additional relevant systems. As such, we spent considerable time validating our original findings using authentic HIV-1 infection in human monocyte-derived macrophages (MDMs), along with testing previously reported Gag mutants with more precise mutations in the MA domain, as well as pulse-chase and preliminary proteomic analysis to confirm HIV-mediated C99 ubiquitination, as outlined in more detail below.

This is an important study toward the relationship between HIV and amyloidogenic pathway. Current manuscript has been written in a way that 'somewhat assumed' readers are cell biologists who already have acquired expertise in HIV biology, amyloid biology, or both. I would encourage authors to bring presentation to the levels of general audience to enhance its impact. For example, Fig1A from the Naghavi lab 2017 paper (Chai et al 2017 DOI: 10.1038/s41467-017-01795-8) nicely capture the amyloid processing biology and set the tone for their 2017 paper back then. Something like such graphic representation in some of the figures would help message delivery.

We thank the Reviewer for their supportive comments and the suggestion to include illustrations to help readers. We have now added illustrations at appropriate points to capture key processes or models, including introducing the amyloid processing

pathways in Fig. 1a as well as our overall model for how Gag and APP compete for control of vesicular sorting pathways in Fig. 10. We thank the Reviewer for this suggestion.

Current story feels somewhat incomplete for this assessor. Whether it is the need to use more relevant model system (infection [or even single round transduction]) into relevant primary cells, biophysical studies to reveal mechanisms better, and / or more appropriate mutagenesis analyses; are needed (in the view of this assessor) to discern the HIV Gag and amyloid relationships. Selected additional approaches (such as iPC brain organoid, and/or proteomics analyses on pull-down samples) may also enrich the current story.

We appreciate the Reviewer's point and as detailed above, we have now added a substantial amount of new data confirming our key findings using multiple independent approaches such as authentic infection in MDMs as well as testing the effects of MA mutants developed by the Freed lab. We also show preliminary proteomic analysis of pull-down samples, below. We respectfully feel that iPC brain organoids and biophysical assays are not sufficiently developed to address the questions in hand and are beyond the scope of this report, and we hope that the extensive new data that we have added addresses the Reviewer's main concern. To avoid redundancy, we describe these new data in more detail in response to each specific point raised by the Reviewer below.

I really would like to emphasise that the story being presented is sound, and likely to be a great one. I just feel that perhaps some of the 'I's and 'T's need to be dotted and crossed with more rigorous data and interpretations.

We thank the Reviewer for their kind and supportive comments, and we hope that our revisions have sufficiently dotted the 'I's and crossed the 'T's in our story.

- Intracellular vs extracellular budding of HIV in macrophages has been a major debate. In these prior works, multiple studies use primary macrophages. One clear message would be that fluorescent imaging has its limitation on discerning intracellular budding (such those in VCC) vs extracellular release. With the current data presented by authors, perhaps it is more appropriate to call them C63 association but stay away from VCC targeting. In particular, both 293 based or CHEM3 based systems have limitations.

We appreciate the Reviewer's point and we now specifically refer to CD63-positive vesicles rather than VCC's as suggested. We have also confirmed these findings in primary human macrophages, but we agree that it may still be prudent to refer to these sites as CD63-positive vesicles in general, while simply informing readers that these are often considered to be VCC's.

- Works of 2017 (Chai et al 2017 DOI: 10.1038/s41467-017-01795-8) from the Naghavi lab have mapped the involvement of MA aa aa77-111 within the HIV Gag in amyloid biology. Interestingly, early work from the Freed lab back in 2000, Ono et al (<https://doi.org/10.1128/JVI.74.6.2855-2866.2000>) reported that mutations between

residues 84 and 88 of the matrix (MA) domain of HIV-1 Gag cause a retargeting of virus particle formation to an intracellular site. The overlapping of Gag regions between these biology perhaps should be considered / discussed by authors more extensively, rather than a general single group statement in line 330 page 16 that is amongst 20 references.

We thank the Reviewer for this insightful suggestion which has further solidified our findings. Transfecting 293A with the pNL4.3 containing WT or mutant Gag (in residues between 84 and 88 in the matrix, namely 85YG and 87VE) from Freed lab (Ono et al, 2000) we now show that in contrast to WT Gag, 85YG or 87VE fail to reduce endogenous APP or CTFs in these cells. These new findings, presented in Fig 1e, nicely complement our prior work using larger MA mutants deleted across residues 72-111 in amyloid biology and combined, provides further evidence that the MA domain influences Gag membrane usage and its effects on APP processing, in line with our broader model of competition between Gag and APP that affects their vesicle sorting and processing. These new findings have also been discussed more extensively in the discussion section of the revised manuscript.

- Primary blood- (or monocytes-) derived macrophages and microglia (brain macrophages) are going to be different from 293 and CHEM3 cells in many major ways. In practice, blood cells derived macrophages are much easier to study than isolating resident microglia cell lines (in my view). As abstract described 'HIV infection in macrophages and microglia (3rd sentence). I was left with impression (and expectation that authors will provide data on 'infection of macrophages with brain isolates, such and JR-CSF'. However, there is no data on HIV INFECTION of blood macrophages as I can see. Perhaps, validation data from more NATURAL HIV INFECTION of blood macrophages would be a good way to overcome inherited limitation on current transfection system. Should authors not choose to incorporate macrophage INFECTION data, abstract / summary description needs to be much more 'controlled' to reflect the biological data being presented – for example microglia cell line would be appropriate.

We thank the Reviewer for encouraging us to test primary macrophages and we have now added a substantial amount of new data confirming our key findings obtained originally in microglia cell lines, now using authentic HIV-1 infection in MDMs. These include: confirmation of the effect of time course HIV-1 infection on the amyloidogenic processing of APP/CTFs in MDMs (Fig. 1d); confirmation of C99 ubiquitination suppressing Gag entry into CD63-positive vesicles in HIV-1 infected MDMs stably expressing C99 or 7KA mutant as a new figure (Fig. 5); confirmation of Gag localization to CD63-positive vesicles in MDMs infected with HIV-1 (Fig. 6 g-i); and confirmation of findings that APP mediates the effects of gamma-secretase inhibitors on the entry of Gag into CD63-positive MVBs (Fig. 8).

- Data in Figure 1 is OK. However, is transfection the 'right' system to ask question on biology of HIV in CNS resident cells? I seem to recall expression of HIV gag in microglia

cells is not high to begin with Thereby, transfection system will largely skew the effect via over-expression.

We apologize if this was due to a lack of clarity on our part, but the Reviewer seems to have overlooked that the data in Figure 1b (now Fig. 1c in the revised manuscript) presents HIV-1 infection, not transfection, in the microglia line. We have ensured that this is clearly detailed in the revised manuscript. Moreover, as the Reviewer rightly points out, microglia cells are not as susceptible to HIV-1 infection and that is why we use CHME3 4x4, which is a human microglia line which expresses higher levels of receptor/coreceptor for more efficient infection with WT HIV-1 envelope. Also, we have now confirmed this using WT HIV-1 infection in MDMs (Fig. 1d).

- In line 129, authors stated ‘infection’ of CHME3 4x4. Based on materials and methods, the work described is transfection but not infection. Infection is misleading. Transfection is generally over-expressing plasmid encoded genes.

Line 129 refers to Fig. 1c, which is infection of CHME3 4x4 as correctly stated in the text and the Figure legend. We apologize for this misunderstanding stemming from the lack of clarification in the materials and methods section. We have now added the details for infection of CHME3 4x4 along with the new infection data in primary human macrophages in the materials and methods, under the “viruses and infections” section in the revised manuscript.

- Lines 139-141. This reviewer has reservation on the claim that HIV Gag accelerate pathway of APP cleavage. Based on current data, it appears to be a conjecture. Pulse-Chase experiments can however demonstrate this claim nicely and directly.

We thank the Reviewer for their insightful suggestion. We have now added new data showing a reduction in both APP and CTFs with increasing amount of HA-tagged Gag, but not the control GAPDH-HA, in the presence of cycloheximide (CHX). This new data demonstrating increased turnover is presented in Fig. 1h. in the revised manuscript.

- Given it is difficult to demonstrate HIV expression in microglia cells in vivo, it is ok to use tissue culture system. Having said that, a more appropriate model (or validation) might be using lentiviral transduction system to control the (titre) gene delivery at 1 copy of gene to be delivered per cell.

We appreciate the Reviewer’s point but we respectfully feel that limitations to detecting HIV-1 infection in vivo are likely as much to do with limitations to in vivo approaches, and that our in vitro systems are appropriate, in particular now that we have added extensive new data using authentic lentivirus infection of MDMs thanks to the Reviewer’s helpful suggestions above. In addition, we feel that dropping the gene delivery to such a low level may not be a true reflection of in vivo infection, where it is unlikely that just one infectious particle enters the cell particularly during productive infection, and also reduces the sensitivity of in vitro assays to measure effects on various processes.

- Legend in Fig 1 said 'C99'. Was there specific information on Ab to indicate it was not C83? A descriptive sentence on 6E10 would be helpful for non-APP readers.

We apologize for the typo in regards to the antibody used. The antibody used here, as correctly referred to in the material and methods section was Y188, not 6E10 as mistakenly referred to in the main text. Available APP antibodies including Y188 do not reliably distinguish between CTF products and that is the main reason we generated flag-tagged versions of both CTFs. We have corrected the term "C99" to "CTFs" in the legend for Fig. 1 as the Reviewer has correctly pointed out. We apologize for this oversight and thank the reviewer for spotting this.

- Legend in Fig1e-f used the word 'uninfected cells', it will be important to use the 'word un-transfected' instead. While current description is 'correct' but can be a bit misleading given the data in Fig1 is transfection system.

We have corrected this in the legend in Fig. 1i-j (Fig1e-f in the original manuscript) in the revised manuscript.

- Not exactly sure the value of the 'time course' in figure 1. First, there is no obvious differences with data between time points. Second, these are transfection data, the system is already overwhelmed with 'over-expression' from plasmid in comparison with natural infection processes. Perhaps remove time point data in Fig 1.

As mentioned above, only data in Fig. 1b is by transfection. Data shown in Fig. 1c and the new Fig. 1d are both by infection in CHME3 4x4 and primary macrophages, respectively. The 2 time points in 293A cells in Fig. 1b were included to simply show consistency in the effect at different timepoints, while the time course is shown for natural infection process at day 5 and 7 in primary macrophages. We respectfully feel that this data is important to show.

- Fig2a-b onFlag-C99 vs Flag-C83 are nice beginning – I would thought that authors could have taken advantage of large body of accumulated Gag point mutants (with known biology) in public domain to evaluate how these HIV Gag would have impacted on amyloid processing (non-amyloidgenic- vs non-amyloidgenic-pathways). More specifically, rather than introducing FlagC-99 and Flag-C83 separately, would it not be more informative to introduce N-terminal tagged FLAG-APP and C-terminal tagged APP-HA in parallel experiments to perform pulse chase in the presence vs absence of HIV Gag – would it not be more informative to show the dynamic of the APP processing starting from full-length APP?

We thank the Reviewer for these suggestions. With regard to the existing Gag mutants, as discussed above, we have tested two overlapping MA mutations in the context of full-length HIV-1 (85YG and 87VE) from Freed lab (Ono et al, 2000). In line with our prior work using quite large deletion mutants, our findings demonstrate that mutations in the MA domain impair HIV-1's ability to increase APP processing. This broadly supports the

idea that WT HIV-1 Gag intersects with APP through its MA domain, and that mutations in MA can redirect and separate the sorting and processing of Gag and APP.

In terms of using N-terminal tagged FLAG-APP and C-terminal tagged APP-Myc, we found that dual tagging affected APP processing by HIV-1. Given that HIV-1 infection or Gag expression clearly increase processing of endogenous APP or singularly tagged APP, we suspect that the dual tagging strategy may affect interactions with Gag or other processes that prevent us from performing these experiments. We hope to explore the underlying reasons for this in the future, but we are likely exploring effects of multi-tagging APP that are not relevant to natural processing. As such, we opted to spend our time on the MA mutants above as well as the broader question of authentic HIV-1 infection in MDMs.

- Pharmacological agents (to me) are helpful with mechanism dissection, but they have issues (in my view) of pleiotropic effects – I consider supportive information; hence I would prefer to see additional experimental approaches.

We appreciate the Reviewer's point but with respect, we use multiple pharmacological agents to rule out such effects and because they are easier to work with in more complex systems. However, beyond these agents, we also use genetic approaches such as C99 mutants to validate data obtained with TAK243. We have also added additional approaches such as UBE1 knockdown to further support these findings; please also see response to the comment below.

- Mutations in Fig 2f-g can only illustrate ala mutations led to reduction of protein levels, but NOT NECESSARY via ubiquitination.

We have now added new data using multiple independent approaches that support the idea that C99 is ubiquitinated. Combined, we now show that treatment with MG132 or TAK243, mutation of key lysines in C99, or depletion of a key regulator of the ubiquitination pathway, UBE1, blocks the reduction of Flag-C99 induced by JR-CSF in 293A cells. These new data are presented in Fig. 2e-f and 2g-h, respectively in the revised manuscript. Moreover, as noted by the Reviewer below, we also detect ubiquitination via Western Blotting analysis. While we appreciate that it is possible that other modifications, such as NEDDylation might also be involved, we respectfully feel that this data is sufficient to support our main conclusion that ubiquitination is indeed involved.

In addition, our preliminary data using pull-down/mass spectrometry further supports this idea by demonstrating that C99, but not the 7KA mutant is ubiquitinated in JR-CSF transfected HEK293A cells. We show this data below for the Reviewer's consideration.

However, this is a complex approach that will require considerable time to optimize in order to assess multi-site ubiquitination, yet it will add little to our current report given the multiple independent approaches used. As such, we feel that although it further supports our conclusions this is beyond the scope of the current report.

- Also, there is an alignment issues for western data in Fig 2f and the densitometry data in Fig 2g. There are 18 samples in Fig 2f, but only 16 densitometry analyses in Fig2g. I suspect these are related to the mock b-CTF transfection control lanes (in the first) two. Readers however should not have to guess, correction on these would be important.

We thank the Reviewer for spotting this alignment error which was indeed caused for the reason suggested. We apologize for this oversight and we have corrected the alignment in Fig 2f-g (now Fig. 2j-k) in the revised manuscript.

- With Fig2h that authors already have the immunoblots on ubiquitination, I do not agree with authors' comments on Flag-7KA does not have ubiquitination. Different levels of ubiquitination between FLAG-C99 and FLAG-7KA sure, but no ubiquitination as stated in legend would be 'incorrect' for me. Given these samples are ready, perhaps it would be wise to cut out these bands from PAGE to perform a target proteomics on C99 WT and mutants to get the profile. Commercially available proteomic quantification kits may also be used to measure potential differences on levels on post-translational modifications.

With respect, the bands detected in Flag-7KA mutant IP's look the same as background in controls. However, we do appreciate the Reviewer's point that there could be a very low level still occurring and as such, we have softened the tone of this statement to say that ubiquitination was "not readily detectable above background levels".

As shown above, this is further supported by preliminary pulldown-mass spectrometry analysis, as suggested by the Reviewer. However, this is not as straightforward as it might seem as there are no commercial proteomic quantification kits available to determine the level of protein ubiquitination. Maybe the Reviewer is referring to the diGly antibodies from Cell Signaling, but our collaborators in the Savas Lab have substantial experience in using this reagent (PMID: 33238128) and have found it to be rather unreliable. As detailed above, we provide preliminary MS/MS data documenting C99 ubiquitination and although too preliminary and technically complex to pursue for this current study, it does further support the multiple independent lines of evidence that we do present in the manuscript that C99 is ubiquitinated.

- Associations amongst Gag, C99, Rab7, and CD63 in Figs 3-4 are good data. Claims on 'Preventing C99 ubiquitination' would be overstated in my view. C-99-7KA has its limitation. One will need quantitative proteomics first to make such statement. 293 and CHME3 cells have their limitations – primary cell system would have been better.

Please see our responses to these points in earlier responses to the same comments.

- Just want to be clear – I am comfortable with the reported effects authors made with C-99-7KA. I am just less confident that C-99-7KA is clean enough to be used as ubiquitination negative situation.

We thank the Reviewer for their supportive stance towards this approach and we appreciate, and have toned down, our ubiquitination statements with regard to this mutant. Please see our responses in regards to C99-7KA ubiquitination above.

- Also, if the argument is that introducing these protein fragments to alter neuropathology, would it not be more critical to evaluate the impacts of processing of endogenous amyloid precursor proteins?

We respectfully point out that the original Fig. 5a-f (now Fig 6a-f in the revised manuscript) shows that prevention of endogenous APP processing using γ -secretase inhibitor limits Gag entry into MVBs, while Fig. 6 (Now Fig. 7) shows that this γ -secretase -mediated inhibition of HIV Gag entry into CD63-positive MVBs requires endogenous APP. We have also now added new data confirming these findings in primary human macrophages in Fig 6g-i and 8 in the revised manuscript. We respectfully feel that these findings address the central comment centered around the focus of this report, namely how APP processing impacts HIV-1's use of intracellular vesicles, while the broader comment regarding neuropathology was the focus of our prior study (Chai et al, Nat. Comm., 2017).

- siRNA study is nice – I would like to see that in relevant system though – primary macrophages for example.

We thank the Reviewer for suggesting this experiment and in addition to expression of the C99-7KA mutant, discussed above, we have also added new data showing that the block to HIV-1 Gag entry into CD63-positive MVBs that is mediated by γ -secretase inhibition requires APP in primary macrophages (Fig 8 in the revised manuscript), similar to our findings in CHME3 cells (Fig. 7).

- Data in Fig5 on gamma-secretase inhibitors are nice to show significant difference at the imaging level – encouraging. However, given the limitation of 'over-expression system' in 'less relevant cell line systems', this assessor will have to reserve judgement. It would have been nice transduction system (at low MOI of 1 to reflect most infected macrophages are likely to have one inserted copy of HIV) into primary macrophages (better yet iPC microglia cells). As it is 'presumed' that HIV infected microglial cells are derived from transmigration of infected monocytes across the BBB, the 'simpler version' (than real microglial cells) of blood monocytes derived macrophages should be sufficient in demonstrating this claim by the authors in a setting that is closer aligned to the situation in vivo.

We have now added confirmation of these initial findings obtained using JR-CSF transfection in HEK293A and CHME3 cell lines in MDMs infected with WT HIV in the presence and absence of γ -secretase inhibitor. These new findings are presented in Fig 6g-i in the revised manuscript.

Reviewer #2 (Remarks to the Author):

Amyloid precursor protein (APP) has been recently identified as an inhibitor of HIV-1 infection in macrophages and microglia but the underlying mechanism remains unknown. Here the authors reveal that entry of HIV-1 Gag into VCCs is blocked by the C-terminal “C99” product that is generated by amyloidogenic processing of APP, but not by the non-amyloidogenic product “C83”. To counter this, Gag promotes multi-site ubiquitination of C99 which controls both exocytic sorting of MVBs and further processing of C99 into toxic amyloids. Processing of C99, entry of Gag into MVBs and release of infectious virus could all be suppressed by expressing ubiquitination-defective C99 or by gamma-secretase inhibitors. The experiments are well executed, provide clear results and display appropriate controls. The findings are interesting as they suggest a mechanism that might suppress the intracellular replication of HIV-1 in brain-resident microglia. However, given that the relevance of the findings is connected with the possibility of treating HIV-associated dementia (HAND), the manuscript does not provide evidence that this strategy might be really beneficial against HAND, as no animal model of HAND is tested in the experimental design of the manuscript. I would consider the whole manuscript as a preliminary study in support of the possibility of using the strategy in vertebrate systems.

We thank the Reviewer for their positive and supportive comments about our overall findings. We appreciate their central concern but respectfully point out that our prior study addressed this as best as is possible, using neuronal co-culture models (Chai et al, Nat. Comm., 2017). The current report focuses more on understanding precisely how APP affects virus replication in relevant macrophages and microglia cells, and how HIV-induced ubiquitination overcomes this restriction. We respectfully feel that this is a significant advance in our understanding of the overall role of APP in limiting HIV-1 infection and, combined with our prior study, provides detailed insights into both sides of this phenomenon in terms of the mechanistic basis of restriction and evasion, along with the potential consequences for HAND development and treatment, as outlined in our discussion. However, taking this beyond in vitro models is impossible for several key reasons; HIV-1 does not infect rodents and primate models using SIV are extremely expensive, while it remains unclear if they recapitulate HAND and even if they do, it takes decades to actually develop HAND, making it extremely hard to directly model and test.

Reviewer #3 (Remarks to the Author):

This manuscript extends Dr. Moghavi’s earlier work on the interactions between HIV-1 Gag protein and amyloid precursor protein (and its fragments) and the implications of these interactions for HIV-induced neurological disease. The manuscript presents extensive data supporting the hypothesis that the APP fragment C99 interferes with the trafficking of HIV Gag into multivesicular bodies, and that Gag in turn drives the ubiquitination and resulting degradation of C99.

In general I found the manuscript clear and convincing, and a useful addition to the literature on the important problem of how HIV infection induces neurological disease. We thank the Reviewer for their kind and supportive comments about our work and we thank them for their suggestions below.

There are a few minor issues that should be clarified.

1. I don't see any mention of the nature or source of the 6E10 monoclonal antibody.

As detailed in response to a similar comment from Reviewer 1, we apologize for the typo with regards to the APP antibody used. The antibodies used were either Y188 (detects both APP and CTFs) or LN27 (detects only APP, not CTFs), not 6E10, as mistakenly referred to in the text. Y188 is from Abcam (ab32136) and LN27 from Invitrogen (130200) and we have included this information and elaborated on their nature in the material and methods section.

2. The Westerns in Fig. 7 have sections labeled "Pr55 Gag S.E." and "L.E." After much head-scratching I decided these must be Short Exposure and Long Exposure. But that doesn't seem right either, as the bands in left half of panel a S.E. are much darker than those in the right half of panel a S.E., but they seem equivalent in the L.E. set. Some explanation would certainly help.

We apologize for any confusion caused. The Reviewer is correct that "S.E." and "L.E." stand for short exposure and long exposure. The reason the bands are different in the short exposure but seem the same in the long exposure is because they become saturated in the longer exposures that are needed to bring up other bands, such as processed p24. We have clarified this in the figure legend of Fig. 9a (original Fig. 7a) to explain this more clearly to readers.

3. In the description of the production of pQCXIP vector particles, the list of plasmids used includes "pCMV-intron" (line 482). What intron and for what purpose? Is this an insert accompanying some coding region in a plasmid?

We apologize for the lack of information about this commonly used vector. We have now specified that pCMV-intron is an MLV based Gag-Pol expressing vector in the material and methods section. This is a CMV-promoter expression plasmid containing intron A from the hCMV immediate-early gene which has been reported to improve transgene expression in different mammalian cell lines and as such is commonly used in expression plasmids (Chapman BS et al., 1991. PMID: 1650459)(Mariati HO et al. 2010. PMID: 19899222).

REVIEWERS' COMMENTS

Reviewer #1 (Remarks to the Author):

Overall, authors have done a very respectable job to revise the manuscript. There are sufficient data in current manuscript that warrant publication in Nature Communication. This assessor has a different opinion on whether HIV assemble and bud into intracellular compartments in macrophages or not. However, such different opinion should always be independent from my support of current manuscript based on quality data. Wordings in revised manuscript are more controlled and measured, which I appreciate.

While the pathology of HAND can only be evaluated in vivo (as highlighted by others) and I would (also) like to see them, YET these 'limitations' should not prevent current manuscript to be published.

I would however encourage the authors to consider a number of minor suggestions below to ensure the inclusion of a wider view of HIV biology, yet without compromising the views of the authors.

- Perhaps title should be 'HIV-1 PROMOTES ubiquitination of the amyloidogenic C-terminal fragment of APP to prevent inhibition of its vesicular replication'. Afterall, HIV does not perform the act of ubiquitination.
- With line 58 on page 3, 'These intracellular localisation and assembly phenotypes' perhaps should more appropriately described as These intracellular / intraluminal localisation and assembly phenotypes. This reviewer does not want to suppress the opinions of the authors but feels a balance approach in description to account a significant opinion of the field is warrant.
- With line 60 on page3, similar comment on 'intracellular assembly' should be rephrased as 'intracellular / intraluminal assembly'. This reviewer does not want to suppress the opinions of the authors but feels a balance approach in description to account a significant opinion of the field is warrant.
- With lines 222-223 on page 11, the subheading 'Preventing C99 ubiquitination suppresses HIV-1 Gag entry into CD63-positive vesicles' should perhaps be more appropriately be referred to as 'Preventing C99 ubiquitination suppresses HIV-1 Gag entry into CD63-positive compartments / vesicles. The likelihood for some of these CD63 associated regions are extracellular cannot be completely ruled out.

Reviewer #3 (Remarks to the Author):

This manuscript extends Dr. Moghavi's earlier work on the interactions between HIV-1 Gag protein and amyloid precursor protein (and its fragments) and the implications of these interactions for HIV-induced neurological disease. The manuscript presents extensive data supporting the hypothesis that the APP fragment C99 interferes with the trafficking of HIV Gag into multivesicular bodies, and that Gag in turn drives the ubiquitination and subsequent degradation of C99. In general I found the manuscript clear and convincing, and a useful addition to the literature on the important problem of how HIV infection induces neurological disease. The minor questions in my review of the original version of the manuscript have now been satisfactorily addressed.

REVIEWER COMMENTS

Reviewer #1 (Remarks to the Author):

Overall, authors have done a very respectable job to revise the manuscript. There are sufficient data in current manuscript that warrant publication in Nature Communication. This assessor has a different opinion on whether HIV assemble and bud into intracellular compartments in macrophages or not. However, such different opinion should always be independent from my support of current manuscript based on quality data. Wordings in revised manuscript are more controlled and measured, which I appreciate.

While the pathology of HAND can only be evaluated in vivo (as highlighted by others) and I would (also) like to see them, YET these 'limitations' should not prevent current manuscript to be published.

I would however encourage the authors to consider a number of minor suggestions below to ensure the inclusion of a wider view of HIV biology, yet without compromising the views of the authors.

We thank the Reviewer once again for their support of our findings and their fairness in assessing our work based on the quality of data. We hope that our changes to the manuscript and responses below address their remaining minor concerns satisfactorily.

- Perhaps title should be 'HIV-1 PROMOTES ubiquitination of the amyloidogenic C-terminal fragment of APP to prevent inhibition of its vesicular replication'. Afterall, HIV does not perform the act of ubiquitination.

We agree with the Reviewer and have changed the title as suggested.

- With line 58 on page 3, 'These intracellular localisation and assembly phenotypes' perhaps should more appropriately described as These intracellular / intraluminal localisation and assembly phenotypes. This reviewer does not want to suppress the opinions of the authors but feels a balance approach in description to account a significant opinion of the field is warrant.

The wording has been changed as suggested.

- With line 60 on page3, similar comment on 'intracellular assembly' should be rephrased as 'intracellular / intraluminal assembly'. This reviewer does not want to suppress the opinions of the authors but feels a balance approach in description to account a significant opinion of the field is warrant.

The wording has been changed as suggested.

- With lines 222-223 on page 11, the subheading 'Preventing C99 ubiquitination suppresses HIV-1 Gag entry into CD63-positive vesicles' should perhaps be more appropriately be referred to as 'Preventing C99 ubiquitination suppresses HIV-1 Gag entry into CD63-positive compartments / vesicles. The likelihood for some of these CD63 associated regions are extracellular cannot be completely ruled out.

The wording has been changed as suggested.

Reviewer #3 (Remarks to the Author):

This manuscript extends Dr. Moghavi's earlier work on the interactions between HIV-1 Gag protein and amyloid precursor protein (and its fragments) and the implications of these interactions for HIV-induced neurological disease. The manuscript presents extensive data supporting the hypothesis that the APP fragment C99 interferes with the trafficking of HIV Gag into multivesicular bodies, and that Gag in turn drives the ubiquitination and subsequent degradation of C99.

In general I found the manuscript clear and convincing, and a useful addition to the literature on the important problem of how HIV infection induces neurological disease. The minor questions in my review of the original version of the manuscript have now been satisfactorily addressed.

We thank the Reviewer again for their kind words and support of our work.